# Learning Brain Representation with Hierarchical Visual Embeddings

**Jiawen Zheng**[1]*, **Haonan Jia**[2]*, **Ming Li**[3], **Yuhui Zheng**[1], **Yufeng Zeng**[1], **Yang Gao**[2], **Chen Liang**[1]†

[1]The Hong Kong University of Science and Technology (Guangzhou)
[2]Beihang University    [3]Tsinghua University
`jzhengbx@connect.hkust-gz.edu.cn, chenliang2@hkust-gz.edu.cn`

## Abstract

Decoding visual representations from brain signals has attracted significant attention in both neuroscience and artificial intelligence. However, the degree to which brain signals truly encode visual information remains unclear. Current visual decoding approaches explore various brain–image alignment strategies, yet most emphasize high-level semantic features while neglecting pixel-level details, thereby limiting our understanding of the human visual system. In this paper, we propose a brain–image alignment strategy that leverages multiple pre-trained visual encoders with distinct inductive biases to capture hierarchical and multiscale visual representations, while employing a contrastive learning objective to achieve effective alignment between brain signals and visual embeddings. Furthermore, we introduce a Fusion Prior, which learns a stable mapping on large-scale visual data and subsequently matches brain features to this pre-trained prior, thereby enhancing distributional consistency across modalities. Extensive quantitative and qualitative experiments demonstrate that our method achieves a favorable balance between retrieval accuracy and reconstruction fidelity.

## 1 Introduction

With the rapid development of text-to-image generative models (Rombach et al., 2022; Zhang et al., 2023; Esser et al., 2024), reconstructing human visual stimuli from brain signals has become a prominent research focus in both neuroscience and artificial intelligence. Visual processing is a core function of the human brain. When visual stimuli is processed by the brain, the primary visual cortex initially deciphers basic pixel attributes such as color, edges, and textures, subsequently forwarding them to various higher-order visual cortices for further hierarchical processing (Blasdel & Lund, 1983; Tsumoto et al., 1978). These higher-level regions collaborate to synthesize and generalize visual data, resulting in semantic characteristics such as objects and environments, and thus formulating the essential processes underlying human visual perception of the external world. (Merigan & Maunsell, 1993).

To investigate these complex and dynamic relationships between the human visual system and brain representations, researchers commonly employ Functional magnetic resonance imaging (fMRI), Magnetoencephalography (MEG), and Electroencephalogram (EEG) for visual decoding and reconstruction (Zhang et al., 2025; Benchetrit et al., 2023). fMRI measures brain activity indirectly through blood-oxygen-level-dependent signals, offering high spatial resolution but limited temporal resolution, which makes it difficult to capture rapid neural dynamics (Logothetis et al., 2001). In contrast, EEG and MEG directly reflect the brain's electrophysiological activity. EEG provides high temporal resolution but suffers from low spatial resolution and a poor signal-to-noise ratio. MEG, while also offering millisecond-level temporal precision, provides comparatively better spatial resolution (Liu et al., 2023a; da Silva, 2013).

Previous research has explored decoding brain signals by aligning them with visual representations, enabling classification, retrieval, and reconstruction. Song et al. (2024) employed contrastive learning to maximize the similarity of matched brain–image pairs while minimizing that of mismatched

---

*Equal technical contribution.
†Corresponding author.

ones. Li et al. (2024) proposed the Adaptive Thinking Mapper (ATM) to align brain signal features with CLIP-derived visual embeddings, combined with a two-stage multi-pipe strategy for brain-to-image generation. However, these approaches rely on direct alignment between brain signals and image features, whereas the structural gap between the two modalities makes this strategy insufficient to capture the underlying shared representations.

Recently, several studies have attempted to improve direct alignment by introducing priors or enriching visual representations. Wu et al. (2025) introduced the Uncertainty-aware Blur Prior (UBP), which mitigates brain–image mismatches by blurring high-frequency image details. Zhang et al. (2025) extended CLIP-derived image embeddings with depth information to enhance brain–image alignment. However, these methods focus primarily on high-level semantic alignment while overlooking low-level pixel information. This oversight prevents a comprehensive understanding of the visual content encoded in brain signals and reduces interpretability.

To bridge the structural gap between the temporal dynamics of brain signals and the hierarchical organization of visual representations, we introduce a Hierarchical Visual Fusion framework with a Fusion Prior, inspired by perceptual mechanisms in the human visual system. The framework integrates multiple pre-trained encoders to construct multiscale visual representations, ranging from pixel-level details to high-level semantics, and leverages contrastive learning to align brain and visual features. To address the limitations of CLIP and related encoders in capturing local and fine-grained information, we incorporate low-level visual features modeled by a Variational Autoencoder (VAE) into the fused representation. This fusion substantially improves zero-shot retrieval performance. In addition, we pretrain a Fusion Prior on large-scale visual data to provide a stable mapping from fused features to diffusion conditions. Aligning brain embeddings with this prior enables faithful image reconstruction and serves as an effective bridge for a brain-to-image decoding framework.

- We propose a fusion-based brain–vision interface that aligns brain embeddings to a fused visual token built from hierarchical encoders (semantics and pixels), together with a pretrained Fusion Prior that bridges this token to a frozen image generation backbone in a reusable, text-free way.

- We provide a key scientific finding: EEG/MEG signals jointly align with high-level semantics and low-level visual details. Within our fusion framework, adding a VAE latent on top of semantic encoders consistently boosts decoding performance, whereas semantics-only or pixels-only settings cannot recover the same brain–vision structure.

- Our method achieves state-of-the-art zero-shot retrieval and improved reconstruction quality, while remaining plug-and-play across different brain encoders under a fixed fusion-based training scheme.

## 2 RELATED WORK

**Brain Visual Decoding** Decoding visual information from brain signals is a primary goal in neuroscience and cognitive computing. Among brain signals, fMRI offers high spatial resolution but limited temporal precision, while EEG and MEG provide millisecond-level temporal resolution. However, MEG requires expensive equipment and controlled environments, whereas EEG is portable but suffers from a low signal-to-noise ratio (SNR). Visual decoding tasks are generally categorized into retrieval and reconstruction. Due to their larger data scale and high temporal resolution, EEG and MEG are commonly adopted for retrieval (Singh et al., 2024; Liu et al., 2025; Song et al., 2024; Du et al., 2023; Wu et al., 2025; Zhang et al., 2025; Li et al., 2024; Wei et al., 2024), while fMRI tends to achieve stronger reconstruction fidelity by capturing detailed spatial information across the brain (Beliy et al., 2019; Gaziv et al., 2022; Takagi & Nishimoto, 2023; Scotti et al., 2023; Lu et al., 2023; Scotti et al., 2024; Fang et al., 2023; Xia et al., 2024). Inspired by these advances, recent studies have explored generating images directly from EEG/MEG signals (Bai et al., 2024; Lan et al., 2023; Lopez et al., 2025; Wei et al., 2024; Li et al., 2024; Fu et al., 2025; Liu et al., 2025), and some have further integrated both paradigms: leveraging retrieval to enhance the generative process (Xie et al., 2024; Zhu et al., 2025). Most existing methods align brain representations with powerful visual priors, such as pixel-level VAE latents (Kingma & Welling, 2013) or semantic CLIP embeddings (Radford et al., 2021), and treat this shared space as the target domain. However, EEG signals contain rich visual information, aligning with a single visual feature cannot

fully capture these details in the brain signals (Takagi & Nishimoto, 2023; Scotti et al., 2023; Singh et al., 2024; Li et al., 2024).

**Cross-modal Contrastive Learning and Hierarchical Visual Representations**  Image-only representation learning increasingly highlights hierarchical structure: self-supervised Vision Transformers capture strong global semantics (Dosovitskiy et al., 2020; Bao et al., 2021; Xie et al., 2022; He et al., 2022; Caron et al., 2021; Oquab et al., 2023), while generative latents such as VAEs/VQ-VAEs provide compact pixel-level codes with high reconstruction fidelity (Kingma & Welling, 2013; Higgins et al., 2017; Van Den Oord et al., 2017; Razavi et al., 2019). Neuroscience evidence further links deeper features to higher visual areas and early layers to fine spatial detail and rapid dynamics (Yamins et al., 2014; Cichy et al., 2016), suggesting that faithful decoding benefits from combining coarse semantics with local cues. In contrast, cross-modal contrastive learning aligns heterogeneous inputs in a single shared space via an InfoNCE-style objective (Oord et al., 2018; Wu et al., 2018; He et al., 2020) and has scaled successfully with bi-encoders for zero-shot transfer (Radford et al., 2021; Jia et al., 2021; Zhai et al., 2022; Li et al., 2022), extending to audio–visual, video–language, and 3D–language settings (Arandjelovic & Zisserman, 2017; Morgado et al., 2021; Wu et al., 2022; Miech et al., 2019; 2020; Xu et al., 2021; Bain et al., 2021; Luo et al., 2022; Xue et al., 2023; Zhang et al., 2022; Liu et al., 2023b). However, such alignment is sensitive to imperfect pairing and domain shift at scale (Jia et al., 2021; Miech et al., 2019), and analyses show a modality gap that hinders fine-grained correspondence (Liang et al., 2022; Wang et al., 2023). In brain visual decoding, where brain–image pairs are scarce and noisy, aligning to a single semantic-only target (e.g., CLIP) can preserve high-level meaning yet underweight low-level textures and spatial details (Cichy et al., 2016; Li et al., 2024; Wu et al., 2025; Zhang et al., 2025). This motivates leveraging hierarchical visual priors (high- and low-level) rather than relying on one shared embedding space.

**Adapters for Image Diffusion Models**  Adapters have emerged as parameter-efficient modules that extend pretrained diffusion models with controllability and editing while largely freezing base weights, offering a unifying recipe across tasks and modalities (Wang et al., 2025). T2I-Adapter (Mou et al., 2024) learns lightweight branches that align external control signals (e.g., edges, depth, sketches) with internal features of a frozen text-to-image model, enabling accurate and composable multi-condition control. ControlNet (Zhang et al., 2023) freezes the pretrained backbone and adds zero-initialized side networks to inject spatial conditions without destabilizing the original prior. IP-Adapter (Ye et al., 2023) decouples cross-attention to integrate image prompts alongside text, delivering strong multimodal conditioning with 22M trainable parameters while keeping the diffusion backbone frozen.

## 3  METHOD

### 3.1  PROBLEM STATEMENT

The goal of visual decoding is to retrieve or reconstruct the visual information corresponding to recorded brain signals. We denote paired brain signals and visual images as $(x_v, x_b) \in \mathcal{D}$, where $x_v \in \mathbb{R}^{H \times W \times 3}$ represents the visual stimulus, with $H$ and $W$ denoting the image height and width, respectively. $x_b \in \mathbb{R}^{C \times T}$ represents the brain signals recorded under the same stimulus, where $C$ corresponds to the number of electrode channels and $T$ indicates the length of time samples.

### 3.2  ALIGNING BRAIN SIGNALS WITH HIERARCHICAL VISUAL REPRESENTATIONS

Directly aligning brain signals with visual representations may fail to capture the intrinsic multiscale nature of the visual information, thereby limiting alignment performance. While high-level semantic features in the visual modality are crucial for category recognition and abstract understanding, low-level features provide complementary structural information and pixel-level details, which are indispensable for improving reconstruction quality. Inspired by this visual perception mechanism (Blasdel & Lund, 1983; Tsumoto et al., 1978), we integrate multiple pretrained visual encoders to separately extract high-level semantic features and low-level pixel features, and align them with brain signal embeddings through a contrastive learning objective to construct a unified hierarchical visual representation.

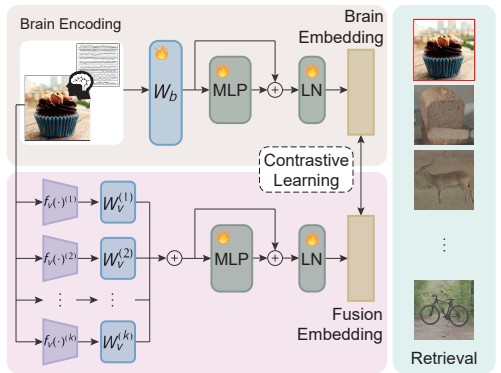
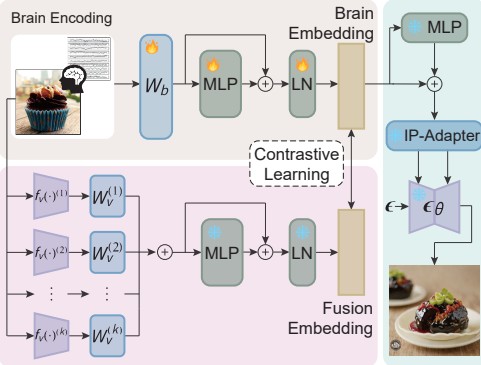

(a) Brain-to-image retrieval.

(b) Brain-to-image reconstruction.

Figure 1: **Learning pipelines**. Left: Retrieval objective that aligns the brain embedding $z_b$ with the fused visual embedding $z_f$ (HVF over $K$ pretrained encoders) using a symmetric InfoNCE; evaluation is nearest-neighbor retrieval in the fused space. Right: Reconstruction pipeline with a frozen, pretrained fusion prior—HVF plus a Conditioning Adapter (MLP projector + IP-Adapter with decoupled cross-attention). We contrastively align $z_b$ to the frozen $z_f$, project to $z_c$, and inject $z_c$ into a frozen SDXL UNet to synthesize the image. Visual encoders and the UNet are frozen; only the brain side is updated during alignment.

**Hierarchical visual representations**   As depicted in Fig. 1-(a), we devise a multi-head encoder structure to obtain hierarchical visual representations ranging from high-level visual semantics (e.g., objects, scenes, and relations) to low-level visual features (e.g., colors, textures, and layouts). We apply $K$ pretrained encoders ($K$=3 by default) to the image $x_v$, yielding $z_v^{(k)} = f_v^{(k)}(x_v)$ for $k$=1,\ldots,K$. For high-level visual semantics, we integrate multiple CLIP encoders and use a single global token from each (i.e., [CLS] token for ViT-based models and the pooled projection for ResNet-based models). For low-level visual features, the VAE encoder outputs a latent of shape $[H/8, W/8, 4]$, which we flatten into a vector of length $(H/8)(W/8) \times 4 = HW/16$, preserving local structure and visual detail.

We fuse features with a post-norm residual Hierarchical Visual Fuser (HVF). For each encoder, a learned linear map $W_v^{(k)} \in \mathbb{R}^{d_k \times d}$ aligns the embedding to the shared dimension $d$=1024:

$$\bar{z}_v = \sum_{k=1}^{K} z_v^{(k)} W_v^{(k)}, \tag{1}$$

The aligned features are fused with a residual Multi-Layer Perceptron (MLP), and we have

$$z_f = \text{LayerNorm}\big(\bar{z}_v + \phi_v(\bar{z}_v)\big), \tag{2}$$

where $\phi_v$ denotes a two-layer MLP with hidden size $d_v = 1024$ and GELU activation.

**Contrastive learning objective**   For the brain modality, we adopt an MLP-based Brain Projection (MBP) network that projects the EEG signal to an embedding. We first flatten the preprocessed signal to $x_b' \in \mathbb{R}^{(C \cdot T)}$, then align it to the visual embedding width using a learned linear projection $W_b \in \mathbb{R}^{CT \times d}$. We then reuse the same architecture as Eq. (2) to produce a $d$-dimensional embedding compatible with $z_f$ with a hidden size of $d$ that

$$\bar{z}_b = x_b' W_b, \qquad z_b = \text{LayerNorm}\big(\bar{z}_b + \phi_b(\bar{z}_b)\big), \tag{3}$$

where $\phi_b$ denotes a two-layer MLP with hidden size $d_b = 1024$ and GELU activation.

We employ a CLIP-style InfoNCE loss (Oord et al., 2018) to align brain and visual embeddings. Given $N$ paired samples, we compute cosine-similarity logits with a trainable temperature $\tau$ (initialized to 0.07):

$$\hat{z}_b^{(i)} = \frac{z_b^{(i)}}{\|z_b^{(i)}\|_2}, \quad \hat{z}_f^{(i)} = \frac{z_f^{(i)}}{\|z_f^{(i)}\|_2}, \quad s_{ij} = \frac{\hat{z}_b^{(i)\top} \hat{z}_f^{(j)}}{\tau}, \tag{4}$$

where $\| \cdot \|_2$ is L2 norm. The learning objective is defined as

$$\mathcal{L}_{\text{contrastive}} = -\frac{1}{2N} \left( \sum_{i=1}^{N} \log \frac{\exp(s_{ii})}{\sum_{j=1}^{N} \exp(s_{ij})} + \sum_{i=1}^{N} \log \frac{\exp(s_{ii})}{\sum_{j=1}^{N} \exp(s_{ji})} \right), \qquad (5)$$

### 3.3 PRETRAINED FUSION PRIOR FOR RECONSTRUCTION

While the above contrastive learning aligns brain signals with hierarchical visual representations, directly feeding these fused representations into a pretrained diffusion model for reconstruction often results in unstable outputs. The core issue is the absence of a stable conditioning prior: brain-driven features do not yet match the distribution expected by the generative model, leading to noisy or misaligned guidance. To address this, we introduce the fusion prior to learn a robust mapping from fused visual features to diffusion conditions.

**Fusion prior pretraining**   As depicted in Fig. 1-(b), we first feed the fused visual representation $z_f$ from the HVF into an additional projector to obtain $z_c$:

$$z_c = z_f + \phi_c(z_f), \qquad (6)$$

where $z_f, z_c \in \mathbb{R}^d$ and both $\phi_v$ in Eq. 2 and $\phi_c$ denotes a two-layer MLP with hidden size $d_c = 4096$ and GELU activation. The IP-Adapter (Ye et al., 2023) then injects $z_c$ into a frozen SDXL (Podell et al., 2023) UNet via cross-attention. Given noisy latent $x_t$ at timestep $t$, the whole network $\delta$ is trained to predict the noise $\epsilon$ with

$$\mathcal{L}_{\text{prior}} = \| \epsilon - \delta(x_t, t, z_c) \|_2^2, \qquad (7)$$

where $\epsilon \sim \mathcal{N}(0, I)$ is the diffusion target and $\mathcal{L}_{\text{prior}}$ is the loss function.

During pretraining on large-scale visual data, the UNet backbone remains frozen, while the HVF and the projector are trained from scratch, the IP-Adapter is initialized from pretrained weights[1] to accelerate convergence. Text prompts are left empty, ensuring the model learns a text-free mapping from fused visual features to diffusion conditions.

**Brain-to-fusion alignment**   Once the HVF is pretrained, we freeze it and update only the brain encoder using the same loss function $\mathcal{L}_{\text{contrastive}}$ as in Eq. (5), which ensures that brain-derived embeddings are projected into a stable, pretrained fusion space. This prevents representational drift and yields robust reconstruction when passed to the diffusion model.

**Full pipeline for reconstruction**   In all, training uses two stages and inference one. (i) *Prior pretraining:* for input images $x_v$, extract $\{z_v^{(k)}\}_{k=1}^{K}$, fuse and project them via the HVF and projector to obtain $z_c$, and train the IP-Adapter jointly with the HVF and projector (UNet frozen) by minimizing $\mathcal{L}_{\text{prior}}$ in Eq. (7) under empty text prompts, yielding a stable, text-free fusion prior. (ii) *Brain–fusion alignment:* freeze the pretrained fusion prior (HVF, projector and IP-Adapter) and the UNet, and update only the brain side (i.e., the MBP module only) on paired $(x_b, x_v)$ with the symmetric InfoNCE loss in Eq. (5) so that $z_b$ lies in the fusion space of $z_f$. (iii) *Reconstruction:* given test brain signals $x_b$, compute $z_b = f_b(x_b)$, feed it to the projector to obtain $z_c$, and use $z_c$ as the sole condition for the frozen IP-Adapter/UNet; a standard diffusion sampler(SDXL uses an Euler–ancestral sampler (Karras et al., 2022)) then produces $\hat{x}_v$, yielding stable and semantically faithful reconstructions.

## 4 EXPERIMENT

### 4.1 EXPERIMENTAL DETAILS

We train the contrastive stage on a single NVIDIA 5090 32GB GPU for 25 epochs with a global batch size of 1024. We use AdamW with a peak learning rate of $5 \times 10^{-4}$ under a cosine decay schedule and a 10-step warmup from zero. Unless otherwise stated, retrieval uses a fixed encoder

---

[1]https://huggingface.co/h94/IP-Adapter/resolve/main/sdxl_models/
ip-adapter_sdxl_vit-h.safetensors

set comprising OpenAI CLIP RN50, LAION CLIP ViT-B/32 (Schuhmann et al., 2022), and an SDXL VAE; each backbone follows its canonical preprocessing. The VAE supports multiple input resolutions and defaults to $128 \times 128$. For generation, we swap RN50 for LAION CLIP ViT-H/14, freeze the pretrained HVF on the visual side, and train only the MBM module of the brain modality. The contrastive stage is trained on THINGS-EEG (Grootswagers et al., 2019; Gifford et al., 2022b)

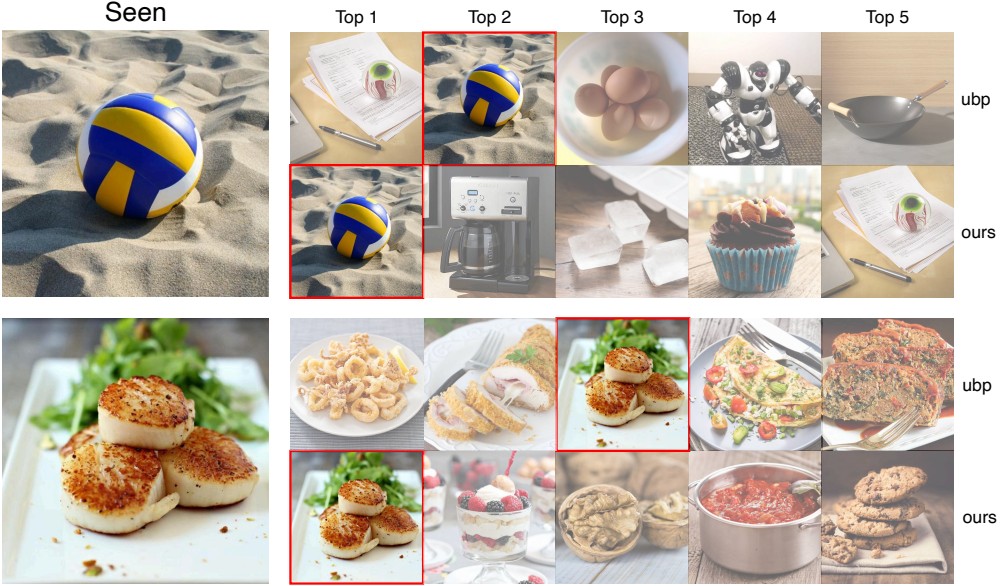

Figure 2: **Hard-case retrieval comparison.** The top-5 retrieved images on the hard-case set from our method and the UBP baseline.

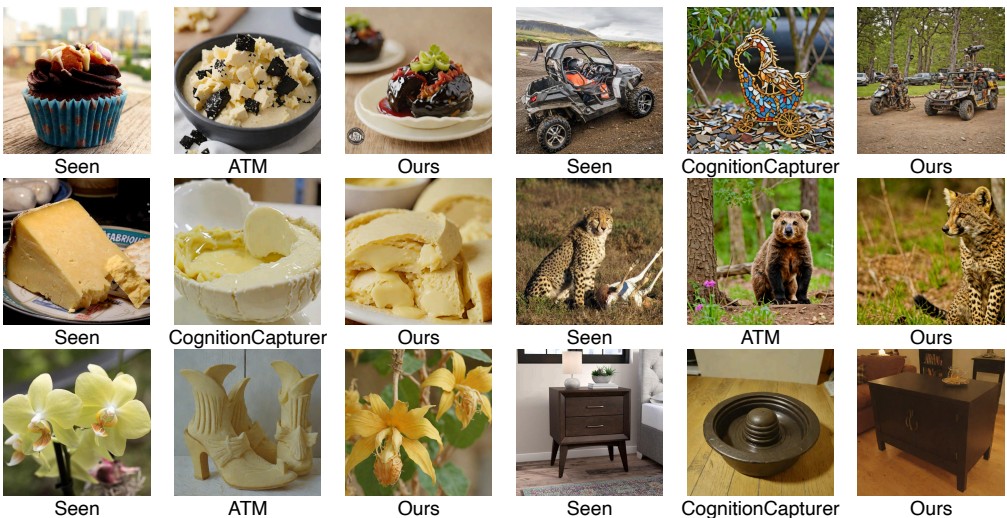

Figure 3: **Qualitative comparison of brain-to-image reconstructions.** Each triplet shows the ground-truth stimulus (left), baseline (middle), and our reconstruction (right). All examples use EEG recordings from subject 8.

and THINGS-MEG (Hebart et al., 2023). For THINGS-EEG (10 participants), the training split contains 1654 concepts with 10 images per concept and 4 repetitions per image; the test split contains 200 concepts with 1 image per concept and 80 repetitions per image. We follow prior work(Li et al., 2024; Wu et al., 2025) to select 17 occipito–parietal channels (O+P) and standard preprocessing (Song et al., 2024). For THINGS-MEG (4 participants, 271 channels), the training split consists of

Table 1: Average Top-1 / Top-5 accuracy (%) for 200-way zero-shot retrieval on THINGS-**EEG** and THINGS-**MEG**. All numbers are subject-wise averages; "–" indicates not reported.

| Method | EEG | | | | MEG | | | |
| --- | --- | --- | --- | --- | --- | --- | --- | --- |
| | Intra-subject | | Inter-subject | | Intra-subject | | Inter-subject | |
| | Top-1 | Top-5 | Top-1 | Top-5 | Top-1 | Top-5 | Top-1 | Top-5 |
| BraVL | 5.8 | 17.5 | 1.8 | 7.0 | – | – | – | – |
| NICE | 16.1 | 43.6 | 6.2 | 21.4 | 12.8 | 36.0 | – | – |
| NICE-SA | 14.7 | 41.7 | 7.0 | 23.1 | 12.7 | 35.0 | – | – |
| NICE-GA | 15.6 | 42.8 | 5.9 | 21.6 | 14.3 | 42.3 | – | – |
| MB2C | 28.5 | 60.4 | 11.9 | 32.0 | – | – | – | – |
| ATM | 28.5 | 60.4 | 11.8 | 33.7 | - | - | – | – |
| VE-SDN | 37.2 | 69.9 | – | – | – | – | – | – |
| CC-All | 35.6 | 80.2 | – | – | – | – | – | – |
| UBP | 50.9 | 79.7 | 12.4 | 33.4 | 26.7 | 55.2 | 2.2 | 10.4 |
| **Ours** | **75.7** | **94.6** | **20.0** | **44.1** | **33.7** | **60.5** | **5.4** | **15.2** |

$1854 \times 12 \times 1$ (concepts × images × reps) and the test split $200 \times 1 \times 12$. To improve signal-to-noise ratio(SNR), repetitions for the same stimulus are averaged within subject in both datasets (training and test). Additional details are provided in the appendix B. For fusion-prior pretraining, we

Table 2: Quantitative assessments of the reconstruction quality for EEG and MEG.

| Method | Dataset | Low-level | | High-level | | | | |
| --- | --- | --- | --- | --- | --- | --- | --- | --- |
| | | PixCorr ↑ | SSIM ↑ | AlexNet(2) ↑ | AlexNet(5) ↑ | Inception ↑ | CLIP ↑ | SwAV ↓ |
| MEG | B.D. | 0.076 | 0.336 | 0.736 | 0.826 | 0.671 | 0.767 | **0.584** |
| | ATM | 0.104 | **0.340** | 0.613 | 0.672 | 0.619 | 0.603 | 0.651 |
| | **Ours** | **0.137** | 0.292 | **0.737** | **0.836** | **0.721** | **0.775** | 0.600 |
| EEG | C.C.(All) | 0.150 | 0.347 | 0.754 | 0.623 | 0.669 | 0.715 | 0.590 |
| | C.C.(Image) | 0.132 | 0.321 | 0.813 | 0.671 | 0.664 | 0.715 | 0.590 |
| | C.C.(Depth) | 0.104 | **0.370** | 0.796 | 0.638 | 0.565 | 0.579 | 0.686 |
| | C.C.(Text) | 0.102 | 0.288 | 0.727 | 0.582 | 0.586 | 0.598 | 0.673 |
| | **Ours** | **0.195** | 0.336 | **0.843** | **0.905** | **0.756** | **0.808** | **0.554** |
| EEG (subj-8) | ATM | 0.160 | 0.345 | 0.776 | 0.866 | 0.734 | 0.786 | 0.582 |
| | **Ours** | **0.227** | **0.361** | **0.878** | **0.924** | **0.796** | **0.826** | **0.531** |

explore multiple prior configurations. Unless noted, training uses two NVIDIA 5090 32 GB GPUs, a fixed learning rate of $1 \times 10^{-4}$, SDXL-base as the diffusion backbone, and the largest feasible batch size of 12 per GPU. Each configuration is trained at $512 \times 512$ for 100k steps, about two epochs, and takes roughly 15 hours per prior configuration. Pretraining uses ImageNet-1k with about 1.3M images. For reconstruction at inference we use SDXL-Turbo with a 4-step sampler for fast evaluation.

## 4.2 QUANTITATIVE EVALUATION

We evaluate two tasks, brain–visual retrieval and brain–visual reconstruction. For retrieval, we report 200-way zero-shot top-1 and top-5 accuracy on THINGS-EEG and THINGS-MEG under both intra-subject and inter-subject protocols. For reconstruction, following prior work (Ozcelik & VanRullen, 2023; Benchetrit et al., 2023; Li et al., 2024), we measure low-level fidelity with PixCorr and SSIM and adopt the remaining semantic and feature-level metrics from these works, including AlexNet(2/5), Inception, CLIP and SwAV distance, where lower is better. The retrieval baselines are BraVL (Du et al., 2023), NICE and its spatial variants (NICE-SA, NICE-GA) (Song et al., 2024), ATM (Li et al., 2024), VE-SDN (Chen et al., 2024), MB2C (Wei et al., 2024), UBP (Wu et al., 2025), and CognitionCapturer (C.C., All/Image/Depth/Text) (Zhang et al., 2025). For reconstruction, we compare with ATM (Li et al., 2024), CognitionCapturer (Zhang et al., 2025), and Brain Decoding (B.D.) (Benchetrit et al., 2023). When prior work reports single-subject results only (e.g., ATM on subj-8), we indicate this in the tables.

Table 3: Ablation on EEG retrieval: average top-1/top-5 accuracy (%) for 200-way zero-shot; we compare single encoders, pairwise, and triple combinations.

| Setting | Configuration | Intra-subject | | Inter-subject | |
|---|---|---|---|---|---|
| | | Top-1 | Top-5 | Top-1 | Top-5 |
| Individual module | B32 | 52.2 | 83.3 | 13.3 | 33.9 |
| | RN50 | 48.1 | 80.4 | 12.7 | 31.7 |
| | VAE | 44.3 | 75.2 | 10.2 | 23.9 |
| Pairwise combination | RN50 + B32 | 56.9 | 86.1 | 14.4 | 36.8 |
| | RN50 + VAE | 65.8 | 90.4 | 17.4 | 37.3 |
| | B32 + VAE | 73.6 | 94.3 | 19.1 | 41.2 |
| Triple combination | RN50 + B32 + VAE | **75.7** | **94.6** | **20.0** | **44.1** |

Compared to the strongest prior work (UBP), our model consistently improves 200-way zero-shot retrieval across all protocols (Top-1/Top-5): EEG intra 75.7/94.6 vs 50.9/79.7, EEG inter 20.0/44.1 vs 12.4/33.4, MEG intra 33.7/60.5 vs 26.7/55.2, and MEG inter 5.4/15.2 vs 2.2/10.4 (Tab. 1). Gains are largest in the inter-subject setting, indicating stronger cross-participant generalization.

Table 2 summarizes reconstruction. On MEG our model matches or exceeds prior work on both low-level similarity and semantic alignment while maintaining a competitive SwAV distance. On EEG it improves the commonly reported subj-8 case and delivers clear subject-averaged gains over ATM and C.C. The average EEG PixCorr increases from 0.150 with C.C.(All) to 0.186 with our model while SSIM remains comparable, and semantic similarities improve across AlexNet, Inception, and CLIP with a lower SwAV distance than C.C. and B.D. Taken together, the metrics indicate that our approach raises both fidelity and semantic agreement and that the improvements persist beyond single-subject evaluation.

## 4.3 VISUAL COMPARISON

In Fig. 2, we show the top-5 retrieved images on the Hard-Case set for our method and the UBP baseline, with our method performing better. We further provide qualitative comparisons with previous brain decoding approaches. As shown in Fig. 3, our method reconstructs images with clearer object contours and more faithful color distribution compared to CognitionCapturer (Zhang et al., 2025) and ATM (Li et al., 2024). In particular, our reconstructions preserve fine-grained structural details while capturing semantically consistent attributes that are often missing in the baselines. Moreover, the overall perceptual quality aligns more closely with the ground-truth stimuli, demonstrating the effectiveness of our framework in bridging brain signals and visual representations.

Table 4: Ablation study on the effect of different fusion priors for Brain-to-Image reconstruction.

| Prior Setting | PixCorr ↑ | SSIM ↑ | AlexNet(2) ↑ | AlexNet(5) ↑ | Inception ↑ | CLIP ↑ | SwAV ↓ |
|---|---|---|---|---|---|---|---|
| H14 | 0.174 | 0.327 | 0.825 | 0.872 | 0.733 | 0.773 | 0.574 |
| H14 + B32 | 0.187 | **0.340** | 0.836 | **0.908** | **0.783** | **0.814** | **0.547** |
| H14 + VAE | 0.173 | 0.312 | 0.789 | 0.838 | 0.672 | 0.721 | 0.611 |
| H14 + B32 + VAE | **0.195** | 0.336 | **0.843** | 0.905 | 0.756 | 0.808 | 0.554 |

## 4.4 ABLATION STUDIES

We ablate the visual encoder composition on 200-way zero-shot EEG retrieval in Tab. 3. Single encoders provide reasonable baselines (e.g., B32: 52.2/83.3 top-1/top-5 intra-subject; 13.3/33.9 inter-subject), and stacking semantic encoders (RN50+B32) yields modest gains (56.9/86.1 intra; 14.4/36.8 inter). Adding the VAE latent gives the largest improvements: B32+VAE reaches 73.6/94.3 (intra) and 19.1/41.2 (inter), and RN50+VAE reaches 65.8/90.4 (intra) and 17.4/37.3 (inter). The full RN50+B32+VAE fusion achieves 75.7/94.6 (intra) and 20.0/44.1 (inter), giving a small but consistent boost over the best pairwise settings on both intra- and inter-subject splits. For brain-

Table 5: Top-1 and top-5 accuracy (%) for 200-way zero-shot retrieval on THINGS-EEG across brain encoder backbones (rows) and visual-encoder combinations (columns), averaged over all 10 subjects and trained with the same fusion-based visual interface and hyperparameters, varying only the brain backbone. *Full* denotes using all three visual encoders (RN50, B32, and VAE).

| | RN50 | | B32 | | VAE | | RN50+B32 | | RN50+VAE | | B32+VAE | | Full | |
|---|---|---|---|---|---|---|---|---|---|---|---|---|---|---|
| Backbone | top-1 | top-5 | top-1 | top-5 | top-1 | top-5 | top-1 | top-5 | top-1 | top-5 | top-1 | top-5 | top-1 | top-5 |
| ShallowNet | 32.4 | 64.9 | 35.7 | 69.9 | 18.5 | 43.4 | 38.2 | 72.3 | 34.4 | 66.6 | 38.7 | 72.1 | 40.8 | 73.9 |
| DeepNet | 17.7 | 44.9 | 18.0 | 46.9 | 6.4 | 20.7 | 19.0 | 48.7 | 11.9 | 36.1 | 17.3 | 43.9 | 17.8 | 45.1 |
| EEGNet | 31.5 | 62.0 | 34.6 | 67.3 | 23.0 | 50.3 | 37.6 | 70.8 | 39.1 | 70.0 | 43.7 | 75.7 | 45.4 | 77.1 |
| TSConv | 38.7 | 72.0 | 42.2 | 74.5 | 25.8 | 55.6 | 45.5 | 78.1 | 46.4 | 78.1 | 52.5 | 83.0 | 54.5 | 84.1 |
| ATM | 41.5 | 74.1 | 42.9 | 76.0 | 34.4 | 63.9 | 46.8 | 78.2 | 52.2 | 81.4 | 56.2 | 85.4 | 57.9 | 86.5 |
| BrainProjection | 47.1 | 78.7 | 50.5 | 82.4 | 43.0 | 74.6 | 55.4 | 85.0 | 64.0 | 89.8 | 72.1 | 93.0 | 72.9 | 94.1 |
| MBP (ours) | **48.1** | **80.5** | **51.2** | **83.0** | **44.1** | **75.5** | **56.2** | **86.2** | **65.2** | **90.7** | **73.7** | **94.1** | **75.7** | **94.6** |

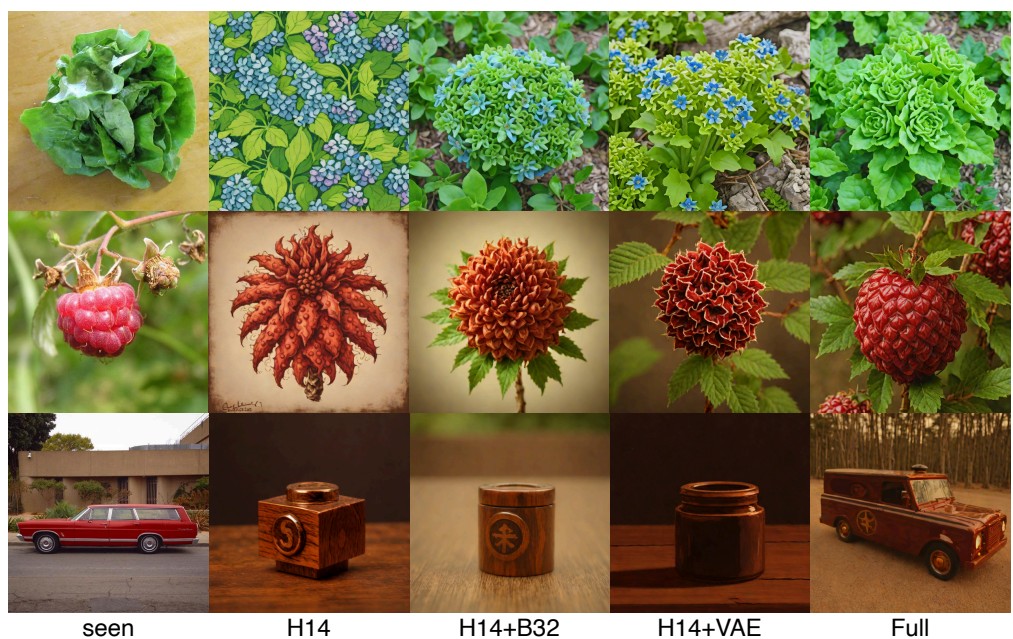

seen · H14 · H14+B32 · H14+VAE · Full

Figure 4: **Ablation on fusion priors.** Each row shows the ground-truth stimulus and reconstructions produced with different fused configurations: H14, H14+B32, H14+VAE, and H14+B32+VAE. All examples use EEG recordings from subject 8.

to-image reconstruction, Tab. 4 compares fusion priors built from H14, B32, and VAE features. Adding B32 to H14 improves both low-level (PixCorr, SSIM) and high-level perceptual metrics, and the full H14+B32+VAE prior further boosts PixCorr and AlexNet similarities while keeping the remaining metrics close to the best two-stream setting; Fig. 4 shows representative qualitative differences between these configurations.

Tab. 5 reports the backbone comparison under the same 17-channel O+P setting and the same fusion-based training. We plug ShallowNet (Schirrmeister et al., 2017), DeepNet (Schirrmeister et al., 2017), EEGNet (Lawhern et al., 2018), TSConv (Song et al., 2024), ATM (Li et al., 2024), Brain-Projection (Wu et al., 2025) and our MBP into the fused visual interface (RN50, B32, VAE and their combinations), keeping the visual encoders, alignment objective, and loss fixed. All prior backbones obtain higher top-1/top-5 retrieval accuracy than their original single-encoder reports, and MBP achieves the highest accuracy in every visual-encoder combination; in the Full RN50+B32+VAE configuration, it reaches 75.7%/94.6% compared to 72.9%/94.1% for BrainProjection, showing plug-and-play use of the same fusion interface across EEG backbones.

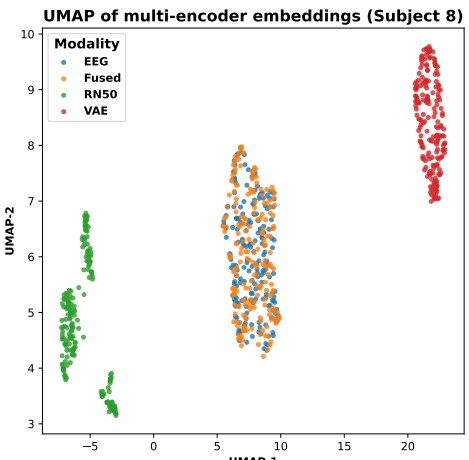 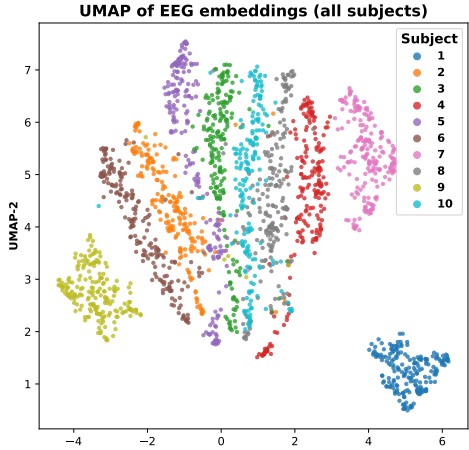

Figure 5: UMAP visualization of learned embeddings on the test split of the THINGS-EEG dataset. Left: multi-encoder visual embeddings (RN50, flattened VAE) and the fused token are projected together with Subject 8 EEG embeddings. Right: EEG embeddings from all 10 subjects.

## 4.5 QUALITATIVE ANALYSIS OF LEARNED EMBEDDINGS

Fig. 5 visualizes the learned embeddings, evaluated on the THINGS-EEG test split after training on the training split. In the left panel, Subject 8 EEG embeddings lie very close to the fused visual tokens and far from both RN50 and VAE embeddings, showing that EEG is aligned to the fused representation rather than to any single encoder. In the right panel, EEG embeddings from the 10 subjects form clearly separated clusters, indicating subject-specific semantic distributions that match the strong inter-subject variability typically observed in EEG. Fig. 7 (in Appendix) exhibits a sharp diagonal between EEG and image embeddings, confirming that the learned fused representation supports reliable 200-way zero-shot retrieval on the test set.

## 5 CONCLUSION

In this paper, we study learning brain representation with hierarchical visual embeddings for brain-to-image decoding. We propose a fusion-based brain–vision interface that aligns brain signals to a single token built from complementary semantic and pixel-level encoders and feeds it into a frozen generation prior. Experiments on THINGS-EEG and THINGS-MEG show that this interface achieves strong 200-way zero-shot retrieval in both intra- and inter-subject settings, together with high-quality reconstructions in intra-subject decoding. Ablation studies show that fusing CLIP and VAE features improves brain-to-image decoding performance over single-encoder and semantics-only baselines. Plugging various EEG backbones into the same interface yields consistent retrieval gains without retraining the visual side, highlighting hierarchical visual embeddings as a plug-and-play route to robust brain representations for brain-to-visual decoding.

**Limitations and future work** Our experiments are restricted to THINGS-EEG and THINGS-MEG; extending the proposed interface to other datasets, modalities, and tasks is an important direction for future work. The current fusion design and visual stack represent a single hand-picked configuration, and exploring stronger vision backbones and alternative encoder sets may further improve decoding performance and efficiency.

## ACKNOWLEDGMENTS

This work was supported by the National Natural Science Foundation of China under Grant No. 62502411.

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

## A LLM Usage Statement

We used LLM for grammar checking and language polishing to improve readability.

## B Datasets Details

**THINGS-EEG** THINGS-EEG (Gifford et al., 2022a) is a large-scale dataset of electroencephalography (EEG) recordings from 10 participants. Signals are acquired with a 64-channel EASYCAP arranged according to the international 10–10 system. The training split spans 1,654 object concepts, each represented by 10 images; every image is shown four times to each participant ($1,654 \times 10 \times 4$). The test split covers 200 concepts with a single image per concept, repeated 80 times ($200 \times 1 \times 80$). Preprocessing follows Song et al. (2024); Wu et al. (2025): raw EEG is filtered to 0.1–100 Hz, yielding 63 channels at 1,000 Hz; trials are segmented from 0–1,000 ms post-stimulus with baseline correction using the prior 200 ms average. Data is then downsampled to 250 Hz, and 17 posterior channels over occipital and parietal sites associated with visual processing are retained. To improve signal-to-noise ratio, repetitions are averaged, producing 16,540 training samples and 200 test samples per participant.

**THINGS-MEG** THINGS-MEG (Hebart et al., 2023) is a large-scale dataset of magnetoencephalography (MEG) recordings from 4 participants. Signals are acquired with 271 channels. Each trial presents an image for 500 ms, followed by a blank screen of $1000 \pm 200$ ms. The training split spans 1,854 object concepts, each represented by 12 images; every image is shown once to each participant ($1,854 \times 12 \times 1$). The test split covers 200 concepts with a single image per concept, repeated 12 times ($200 \times 1 \times 12$). To construct the zero-shot task, 200 test concepts are discarded from the training set. Preprocessing follows Song et al. (2024); Wu et al. (2025): raw MEG is filtered to 0.1–100 Hz; trials are segmented from 0–1,000 ms post-stimulus with baseline correction. Data is then downsampled to 200 Hz. To improve signal-to-noise ratio, repetitions are averaged, producing 19,848 training samples and 200 test samples per participant.

**Ethical considerations** Our experiments rely exclusively on the publicly released THINGS-EEG/MEG datasets (Gifford et al., 2022a; Hebart et al., 2023) collected under informed consent and institutional oversight, and we use only anonymized recordings. While brain-to-image decoding may ultimately benefit populations such as locked-in or speech-impaired patients, it also poses clear risks, including unauthorized inference of mental states and privacy violations. We view our method as a technical contribution on a benchmark dataset, not as a tool for covert monitoring, and we stress that any real-world deployment must require explicit consent, robust data protection, and adherence to clinical and legal regulations.

## C Results Details

**Per-Subject retrieval on THINGS-EEG and THINGS-MEG** We report 200-way zero-shot Top-1/Top-5 accuracy per subject for THINGS-EEG and THINGS-MEG. For each subject, we evaluate individual encoders (RN50, B32, VAE), pairwise stacks (RN50+B32, RN50+VAE, B32+VAE), and the triple stack (RN50+B32+VAE) with the VAE input fixed at $128 \times 128$.

Table 6: Top-1 and Top-5 accuracy (%) for 200-way zero-shot retrieval on THINGS-EEG.

| Method | Sub1 | | Sub2 | | Sub3 | | Sub4 | | Sub5 | | Sub6 | | Sub7 | | Sub8 | | Sub9 | | Sub10 | | Avg | |
|---|---|---|---|---|---|---|---|---|---|---|---|---|---|---|---|---|---|---|---|---|---|---|
| | top-1 | top-5 | top-1 | top-5 | top-1 | top-5 | top-1 | top-5 | top-1 | top-5 | top-1 | top-5 | top-1 | top-5 | top-1 | top-5 | top-1 | top-5 | top-1 | top-5 | top-1 | top-5 |
| BraVL | 6.1 | 17.9 | 4.9 | 14.9 | 5.6 | 17.4 | 5.0 | 15.1 | 4.0 | 13.4 | 6.0 | 18.2 | 6.5 | 20.4 | 8.8 | 23.7 | 4.3 | 14.0 | 7.0 | 19.7 | 5.8 | 17.5 |
| NICE | 13.2 | 39.5 | 13.5 | 40.3 | 14.5 | 42.7 | 20.6 | 52.7 | 10.1 | 31.5 | 16.5 | 44.0 | 17.0 | 42.1 | 22.9 | 56.1 | 15.4 | 41.6 | 17.4 | 45.8 | 16.1 | 43.6 |
| NICE-SA | 13.3 | 40.2 | 12.1 | 36.1 | 15.3 | 39.6 | 15.9 | 49.0 | 9.8 | 34.4 | 14.2 | 42.4 | 17.9 | 43.6 | 18.2 | 50.2 | 14.4 | 38.7 | 16.0 | 42.8 | 14.7 | 41.7 |
| NICE-GA | 15.2 | 40.1 | 13.9 | 40.1 | 14.7 | 42.7 | 17.6 | 48.9 | 9.0 | 29.7 | 16.4 | 44.4 | 14.9 | 43.1 | 20.3 | 52.1 | 14.1 | 39.7 | 19.6 | 46.7 | 15.6 | 42.8 |
| MB2C | 23.7 | 56.3 | 22.7 | 50.5 | 26.3 | 60.2 | 34.8 | 67.0 | 21.3 | 53.0 | 31.0 | 62.3 | 25.0 | 54.8 | 39.0 | 69.3 | 27.5 | 59.3 | 33.2 | 70.8 | 28.5 | 60.4 |
| ATM-S | 25.6 | 60.4 | 22.0 | 54.5 | 25.0 | 62.4 | 31.4 | 60.9 | 12.9 | 43.0 | 21.3 | 51.1 | 30.5 | 61.5 | 38.8 | 72.0 | 34.4 | 51.5 | 29.1 | 63.5 | 28.5 | 60.4 |
| VE-SDN | 32.6 | 63.7 | 34.4 | 69.9 | 38.7 | 73.5 | 39.8 | 72.0 | 29.4 | 58.6 | 34.5 | 68.8 | 34.5 | 68.3 | 49.3 | 79.8 | 39.0 | 69.6 | 39.8 | 75.3 | 37.2 | 69.9 |
| CognitionCapturer-All | 31.4 | 79.7 | 31.4 | 77.8 | 38.2 | 85.7 | 40.4 | 85.8 | 24.4 | 66.3 | 34.8 | 78.8 | 34.7 | 81.0 | 48.1 | 88.6 | 31.4 | 79.4 | 35.6 | 79.3 | 35.6 | 80.2 |
| UBP | 41.2 | 70.5 | 51.2 | 80.9 | 51.2 | 82.0 | 51.1 | 76.9 | 42.2 | 72.8 | 57.5 | 83.5 | 49.0 | 79.9 | 58.6 | 85.8 | 45.1 | 76.2 | 61.5 | 88.2 | 50.9 | 79.7 |
| Ours | 64.3 | 88.8 | 76.3 | 95.3 | 74.0 | 95.0 | 67 | 91.8 | 68.0 | 91.5 | 81.5 | 96.3 | 76.8 | 96.8 | 84.8 | 98.5 | 76.8 | 95.8 | 87.3 | 99.3 | 75.7 | 94.6 |

**Per-Subject reconstruction metrics** We further report reconstruction metrics per subject in Tab. 10 Tab. 11, Tab. 12, and Tab. 13. For each subject, we compute low-level measures (PixCorr,

Table 7: Top-1 and Top-5 accuracy (%) for 200-way zero-shot retrieval on THINGS-EEG across different configurations.

| Configuration | Sub1 | | Sub2 | | Sub3 | | Sub4 | | Sub5 | | Sub6 | | Sub7 | | Sub8 | | Sub9 | | Sub10 | | Avg | |
|---|---|---|---|---|---|---|---|---|---|---|---|---|---|---|---|---|---|---|---|---|---|---|
| | top-1 | top-5 | top-1 | top-5 | top-1 | top-5 | top-1 | top-5 | top-1 | top-5 | top-1 | top-5 | top-1 | top-5 | top-1 | top-5 | top-1 | top-5 | top-1 | top-5 | top-1 | top-5 |
| B32 | 39.3 | 75.3 | 48.8 | 79.3 | 53.3 | 84.5 | 54.8 | 87.0 | 42.8 | 75.0 | 57.8 | 84.3 | 47.0 | 81.0 | 62.3 | 89.3 | 44.3 | 80.0 | 61.8 | 94.3 | 51.2 | 83.0 |
| RN50 | 40.0 | 69.5 | 48.5 | 79.5 | 48.5 | 85.0 | 45.8 | 82.0 | 41.5 | 74.0 | 55.8 | 83.0 | 48.5 | 77.5 | 55.5 | 88.0 | 41.8 | 77.0 | 55.3 | 89.5 | 48.1 | 80.5 |
| VAE | 38.8 | 71.5 | 41.3 | 72.5 | 43.3 | 75.3 | 33.5 | 65.3 | 39.8 | 70.5 | 50.5 | 81.8 | 44.3 | 75.0 | 55.0 | 86.3 | 42.8 | 73.3 | 52.0 | 83.5 | 44.1 | 75.5 |
| RN50+B32 | 47.3 | 79.0 | 55.8 | 81.0 | 56.3 | 87.5 | 59.8 | 88.8 | 46.8 | 81.0 | 63.0 | 87.5 | 53.0 | 85.0 | 65.0 | 91.3 | 50.0 | 86.5 | 68.3 | 94.5 | 56.5 | 86.2 |
| RN50+VAE | 60.8 | 86.0 | 62.8 | 92.0 | 59.8 | 91.0 | 53.3 | 87.0 | 58.0 | 84.0 | 73.0 | 94.0 | 62.5 | 88.8 | 77.0 | 97.3 | 68.3 | 90.8 | 77.0 | 96.5 | 65.2 | 90.7 |
| B32+VAE | 63.0 | 88.3 | 70.3 | 94.0 | 73.5 | 94.3 | 64.3 | 92.3 | 70.5 | 91.0 | 78.0 | 96.0 | 73.3 | 93.3 | 84.8 | 97.5 | 75.0 | 96.0 | 84.8 | 98.8 | 73.7 | 94.1 |
| RN50+B32+VAE | 64.3 | 88.8 | 76.3 | 95.3 | 74.0 | 95.0 | 67 | 91.8 | 68.0 | 91.5 | 81.5 | 96.3 | 76.8 | 96.8 | 84.8 | 98.5 | 76.8 | 95.8 | 87.3 | 99.3 | 75.7 | 94.6 |

Table 8: Top-1 and Top-5 accuracy (%) for 200-way zero-shot retrieval on THINGS-MEG

| Method | Sub1 | | Sub2 | | Sub3 | | Sub4 | | Avg | |
|---|---|---|---|---|---|---|---|---|---|---|
| | top-1 | top-5 | top-1 | top-5 | top-1 | top-5 | top-1 | top-5 | top-1 | top-5 |
| NICE | 9.6 | 27.8 | 18.5 | 47.8 | 14.2 | 41.6 | 9.0 | 26.6 | 12.8 | 36.0 |
| NICE-SA | 9.8 | 27.8 | 18.6 | 46.4 | 10.5 | 38.4 | 11.7 | 27.2 | 12.7 | 35.0 |
| NICE-GA | 8.7 | 30.5 | 21.8 | 56.6 | 16.5 | 49.7 | 10.3 | 32.3 | 14.3 | 42.3 |
| UBP | 15.0 | 38.0 | 46.0 | 80.5 | 27.3 | 59.0 | 18.5 | 43.5 | 26.7 | 55.2 |
| Ours | 14.0 | 31.8 | 63.8 | 91.8 | 41.0 | 78.3 | 17.0 | 41.0 | 33.9 | 60.7 |

SSIM) and high-level perceptual similarity (AlexNet(2/5), Inception, CLIP) with SwAV↓ as a diversity/consistency proxy. Results are shown for the single target (H14 only), semantic pair (H14+B32, H14+VAE) and the full multiscale stack (H14+B32+VAE). The last row gives subject-wise means.

**Reconstruction from different subjects** As shown in Fig. 6, for the same visual stimulus, we reconstruct images from EEG recorded from different subjects.

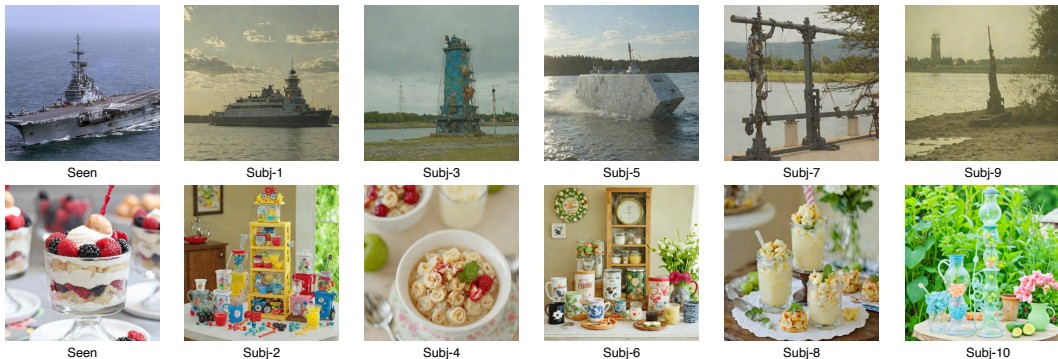

Figure 6: **Cross-subject EEG reconstructions.**

Table 9: Top-1 and Top-5 accuracy (%) for 200-way zero-shot retrieval on THINGS-MEG across different configurations.

| Configuration | Sub1 | | Sub2 | | Sub3 | | Sub4 | | Avg | |
|---|---|---|---|---|---|---|---|---|---|---|
| | top-1 | top-5 | top-1 | top-5 | top-1 | top-5 | top-1 | top-5 | top-1 | top-5 |
| B32 | 9.8 | 31.5 | 52.8 | 81.3 | 31.8 | 67.0 | 19.5 | 47.5 | 28.4 | 56.8 |
| RN50 | 12.0 | 37.5 | 50.3 | 83.0 | 29.8 | 65.8 | 19.0 | 44.5 | 27.8 | 57.7 |
| VAE | 2.9 | 13.2 | 32.5 | 65.3 | 12.4 | 39.1 | 5.6 | 19.1 | 13.4 | 34.2 |
| RN50+B32 | 9.8 | 31.5 | 52.0 | 83.0 | 32.5 | 67.8 | 18.8 | 47.8 | 28.3 | 57.5 |
| RN50+VAE | 9.0 | 22.8 | 48.0 | 85.3 | 26.5 | 61.3 | 11.5 | 30.3 | 23.8 | 49.9 |
| B32+VAE | 12.0 | 33.5 | 64.5 | 91.3 | 39.0 | 76.8 | 17.3 | 43.8 | 33.2 | 61.3 |
| RN50+B32+VAE | 14.0 | 31.8 | 63.8 | 91.8 | 41.0 | 78.3 | 17.0 | 41.0 | 33.9 | 60.7 |

Table 10: Reconstruction metrics across subjects using the H14 setting (higher ↑ is better, lower ↓ is better).

| | Low-level | | High-level | | | | |
|---|---|---|---|---|---|---|---|
| Subject | Pixcorr↑ | SSIM↑ | AlexNet(2)↑ | AlexNet(5)↑ | Inception↑ | CLIP↑ | SwAV↓ |
| 1 | 0.179 | 0.305 | 0.828 | 0.870 | 0.719 | 0.732 | 0.588 |
| 2 | 0.174 | 0.331 | 0.826 | 0.868 | 0.712 | 0.769 | 0.588 |
| 3 | 0.177 | 0.317 | 0.832 | 0.872 | 0.703 | 0.802 | 0.574 |
| 4 | 0.167 | 0.326 | 0.803 | 0.863 | 0.752 | 0.778 | 0.573 |
| 5 | 0.163 | 0.315 | 0.804 | 0.846 | 0.676 | 0.745 | 0.593 |
| 6 | 0.181 | 0.316 | 0.838 | 0.874 | 0.715 | 0.763 | 0.588 |
| 7 | 0.155 | 0.328 | 0.811 | 0.874 | 0.736 | 0.775 | 0.569 |
| 8 | 0.193 | 0.349 | 0.852 | 0.906 | 0.781 | 0.795 | 0.550 |
| 9 | 0.163 | 0.330 | 0.820 | 0.872 | 0.765 | 0.762 | 0.561 |
| 10 | 0.192 | 0.350 | 0.837 | 0.870 | 0.773 | 0.810 | 0.556 |
| Ave | 0.174 | 0.327 | 0.825 | 0.871 | 0.733 | 0.773 | 0.574 |

Table 11: Reconstruction metrics across subjects using the H14+B32 setting (higher ↑ is better, lower ↓ is better).

| | Low-level | | High-level | | | | |
|---|---|---|---|---|---|---|---|
| Subject | Pixcorr↑ | SSIM↑ | AlexNet(2)↑ | AlexNet(5)↑ | Inception↑ | CLIP↑ | SwAV↓ |
| 1 | 0.193 | 0.317 | 0.816 | 0.886 | 0.761 | 0.771 | 0.568 |
| 2 | 0.190 | 0.346 | 0.845 | 0.919 | 0.789 | 0.821 | 0.551 |
| 3 | 0.191 | 0.330 | 0.834 | 0.903 | 0.758 | 0.827 | 0.559 |
| 4 | 0.183 | 0.334 | 0.825 | 0.905 | 0.805 | 0.840 | 0.535 |
| 5 | 0.176 | 0.326 | 0.825 | 0.903 | 0.720 | 0.795 | 0.561 |
| 6 | 0.191 | 0.326 | 0.833 | 0.907 | 0.791 | 0.817 | 0.552 |
| 7 | 0.166 | 0.337 | 0.831 | 0.910 | 0.765 | 0.794 | 0.556 |
| 8 | 0.207 | 0.365 | 0.861 | 0.918 | 0.815 | 0.827 | 0.528 |
| 9 | 0.183 | 0.348 | 0.838 | 0.904 | 0.792 | 0.797 | 0.535 |
| 10 | 0.190 | 0.365 | 0.848 | 0.921 | 0.830 | 0.854 | 0.521 |
| Ave | 0.187 | 0.339 | 0.836 | 0.908 | 0.783 | 0.814 | 0.547 |

Table 12: Reconstruction metrics across subjects using the H14+VAE setting (higher ↑ is better, lower ↓ is better).

| | Low-level | | High-level | | | | |
|---|---|---|---|---|---|---|---|
| **Subject** | Pixcorr↑ | SSIM↑ | AlexNet(2)↑ | AlexNet(5)↑ | Inception↑ | CLIP↑ | SwAV↓ |
| 1 | 0.156 | 0.301 | 0.755 | 0.762 | 0.653 | 0.646 | 0.658 |
| 2 | 0.175 | 0.323 | 0.801 | 0.852 | 0.643 | 0.743 | 0.608 |
| 3 | 0.171 | 0.290 | 0.793 | 0.853 | 0.651 | 0.730 | 0.621 |
| 4 | 0.167 | 0.307 | 0.798 | 0.851 | 0.707 | 0.761 | 0.589 |
| 5 | 0.174 | 0.295 | 0.783 | 0.847 | 0.670 | 0.723 | 0.611 |
| 6 | 0.174 | 0.304 | 0.806 | 0.845 | 0.669 | 0.720 | 0.612 |
| 7 | 0.154 | 0.315 | 0.764 | 0.828 | 0.655 | 0.711 | 0.622 |
| 8 | 0.196 | 0.335 | 0.817 | 0.859 | 0.691 | 0.734 | 0.590 |
| 9 | 0.166 | 0.315 | 0.765 | 0.827 | 0.678 | 0.701 | 0.605 |
| 10 | 0.194 | 0.337 | 0.810 | 0.856 | 0.703 | 0.746 | 0.594 |
| Ave | 0.173 | 0.312 | 0.789 | 0.838 | 0.672 | 0.721 | 0.611 |

Table 13: Reconstruction metrics across subjects using the H14+B32+VAE setting (higher ↑ is better, lower ↓ is better).

| | Low-level | | High-level | | | | |
|---|---|---|---|---|---|---|---|
| **Subject** | Pixcorr↑ | SSIM↑ | AlexNet(2)↑ | AlexNet(5)↑ | Inception↑ | CLIP↑ | SwAV↓ |
| 1 | 0.193 | 0.332 | 0.835 | 0.883 | 0.727 | 0.757 | 0.578 |
| 2 | 0.188 | 0.341 | 0.846 | 0.901 | 0.769 | 0.807 | 0.559 |
| 3 | 0.196 | 0.324 | 0.834 | 0.900 | 0.755 | 0.826 | 0.566 |
| 4 | 0.187 | 0.320 | 0.821 | 0.903 | 0.774 | 0.824 | 0.553 |
| 5 | 0.179 | 0.317 | 0.831 | 0.893 | 0.705 | 0.797 | 0.565 |
| 6 | 0.211 | 0.329 | 0.852 | 0.920 | 0.758 | 0.811 | 0.561 |
| 7 | 0.179 | 0.336 | 0.833 | 0.914 | 0.754 | 0.805 | 0.554 |
| 8 | 0.219 | 0.356 | 0.876 | 0.926 | 0.788 | 0.827 | 0.531 |
| 9 | 0.198 | 0.342 | 0.842 | 0.889 | 0.757 | 0.783 | 0.546 |
| 10 | 0.203 | 0.358 | 0.858 | 0.922 | 0.777 | 0.846 | 0.530 |
| Ave | 0.195 | 0.336 | 0.843 | 0.905 | 0.756 | 0.808 | 0.554 |

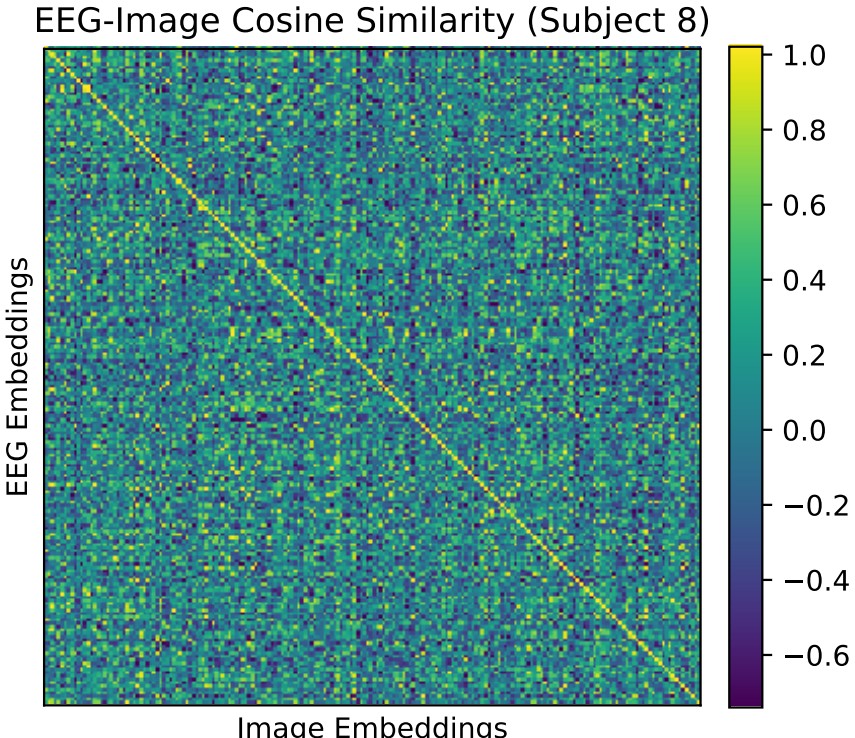

Figure 7: Cosine similarity matrix between EEG and image embeddings for Subject 8 on the test split of the THINGS-EEG dataset. Rows correspond to EEG embeddings and columns to image embeddings; the strong diagonal pattern indicates that each EEG embedding is most similar to its paired image embedding.

## D    ADDITIONAL ABLATION STUDIES

**Extended visual encoder ablations**    Fig. 8 and Tab. 14 compare additional visual encoders (DINO (Oquab et al., 2023) and SynCLR (Tian et al., 2024)) with their multi-encoder variants. Across backbones, adding the SDXL VAE consistently boosts top-1 and top-5 retrieval accuracy in both intra- and inter-subject settings, with especially large gains for DINO. Pairing DINO with CLIP encoders (RN50 or B32) brings moderate improvements, while VAE-based combinations yield the strongest gains, particularly when fused with B32. For SynCLR, stacking with RN50 or B32 alone largely saturates performance and can slightly hurt inter-subject transfer, whereas SynCLR+VAE and VAE-based triple stacks achieve the best overall results. These trends indicate that low-level VAE features systematically complement high-level representations for EEG–image alignment across visual backbones.

**Effect of EEG feature selection on channels**    Unless otherwise stated, we use EEG data from 0–1000 ms after stimulus onset and restrict inputs to 17 occipital–parietal (O+P[2]) channels following previous work (Song et al., 2024). Tab. 15 shows that, for each visual-encoder configuration, the O+P subset consistently attains the highest 200-way retrieval accuracy compared to using only occipital channels, only parietal channels, "other" non-visual channels, or the full 63-channel montage.

**Effect of EEG feature selection on time windows**    Fig. 9 plots top-1 retrieval accuracy as a function of the EEG time window for different visual encoder configurations. For cumulative windows $[0, t]$, accuracy under the full RN50+B32+VAE fusion setting rises from 21.8% at 0–100 ms to 73.9% at 0–300 ms and then stays close to the 0–1000 ms baseline (75.7%), while tail windows $[t, 1000]$ that exclude early activity drop from 75.7% (0–1000 ms) to 69.9% (100–1000 ms)

[2]P7,P5,P3,P1,Pz,P2,P4,P6,P8,PO7,PO3,POz,PO4,PO8,O1,Oz,O2

Table 14: Top-1 and top-5 accuracy (%) for 200-way zero-shot retrieval on THINGS-EEG under extended visual encoder settings. The first two columns report intra-subject performance, and the last two columns report inter-subject performance. Results are averaged over all 10 subjects under the same training configuration, with only the visual encoder stack varied. Numbers in parentheses denote absolute gains over the corresponding single-encoder baseline.

| | Intra-subject | | Inter-subject | |
|---|---|---|---|---|
| Visual setting | top-1 | top-5 | top-1 | top-5 |
| DINO | 23.5 | 50.9 | 10.7 | 22.8 |
| SynCLR | 68.0 | 91.8 | 19.5 | 42.0 |
| DINO+RN50 | 28.6 (+5.1) | 57.7 (+6.8) | 11.9 (+1.2) | 26.3 (+3.5) |
| DINO+B32 | 35.7 (+12.2) | 66.8 (+15.9) | 13.3 (+2.6) | 29.9 (+7.1) |
| DINO+VAE | 50.1 (+26.6) | 79.9 (+29.0) | 16.1 (+5.4) | 34.8 (+12.0) |
| SynCLR+RN50 | 69.1 (+1.1) | 91.9 (+0.1) | 19.2 (-0.3) | 41.0 (-1.0) |
| SynCLR+B32 | 68.4 (+0.4) | 92.4 (+0.6) | 19.2 (-0.3) | 42.3 (+0.3) |
| SynCLR+VAE | 78.5 (+10.5) | 96.4 (+4.6) | 23.0 (+3.5) | 46.5 (+4.5) |
| DINO+RN50+VAE | 53.3 (+29.8) | 82.6 (+31.7) | 17.3 (+6.6) | 35.8 (+13.0) |
| DINO+B32+VAE | 58.5 (+35.0) | 85.9 (+35.0) | 19.3 (+8.6) | 39.8 (+17.0) |
| SynCLR+RN50+VAE | 78.7 (+10.7) | 96.5 (+4.7) | 23.4 (+3.9) | 47.1 (+5.1) |
| SynCLR+B32+VAE | 78.9 (+10.9) | 96.5 (+4.7) | 23.4 (+3.9) | 47.2 (+5.2) |

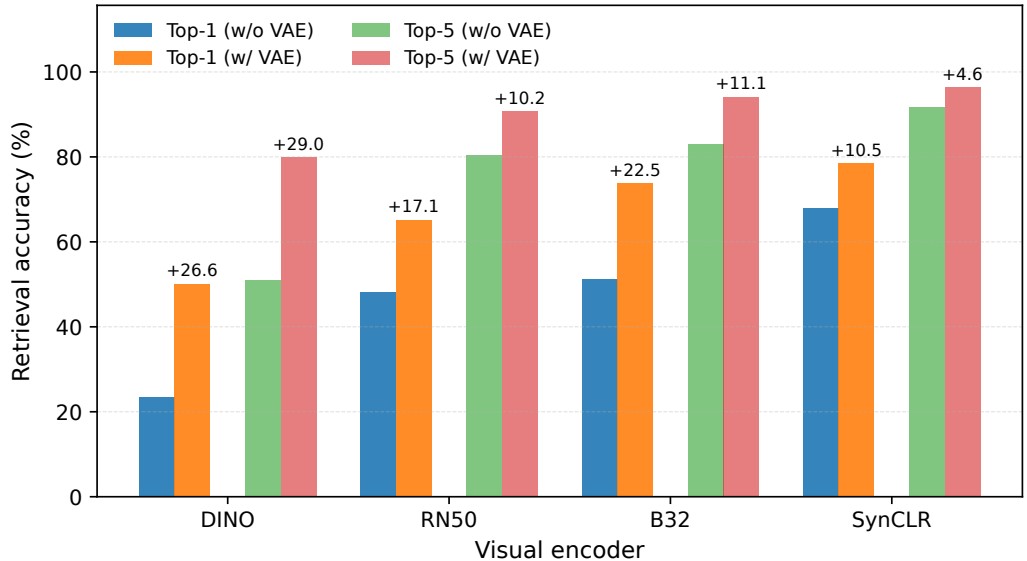

Figure 8: Top-1 and top-5 accuracy (%) for 200-way zero-shot retrieval on THINGS-EEG when aligning EEG to single encoders versus encoder+VAE mixtures. Bars show per-setting performance and absolute gains from adding the SDXL VAE, averaged over all 10 subjects.

and 13.7% (300–1000 ms), approaching chance for 500–1000 ms (1.7%). Comparing encoders, VAE-based configurations tend to outperform CLIP-based ones in the earliest cumulative windows, whereas CLIP-based configurations outperform VAE in later tail windows, suggesting that early EEG activity may be more aligned with low-level, pixel-like features and later activity may be more aligned with higher-level semantic features. For more complete windows (i.e., larger t for [0, t] and smaller t for [t, 1000]), CLIP-dominated and fused configurations generally surpass VAE alone. This is consistent with the possibility that, after a few hundred milliseconds, the EEG signals most useful for 200-way zero-shot retrieval are dominated by hig1her-level visual information. Across all

Table 15: Top-1 and top-5 accuracy (%) for 200-way zero-shot retrieval on THINGS-EEG across visual-encoder settings (rows) and channel selections (columns). Numbers are averaged over all 10 subjects under the same training configuration, with only the selected channels varied.

| Method | Occipital top-1 | Occipital top-5 | Parietal top-1 | Parietal top-5 | O+P (Our) top-1 | O+P (Our) top-5 | Others top-1 | Others top-5 | All top-1 | All top-5 |
|---|---|---|---|---|---|---|---|---|---|---|
| RN50 | 43.1 | 75.51 | 21.8 | 52.1 | 48.1 | 80.5 | 9.4 | 26.9 | 43.1 | 75.7 |
| B32 | 48.3 | 80.0 | 23.9 | 52.8 | 51.2 | 83.0 | 8.5 | 26.4 | 44.1 | 78.3 |
| VAE | 45.9 | 76.6 | 10.0 | 27.7 | 44.1 | 75.5 | 2.8 | 10.9 | 27.0 | 59.0 |
| RN50+B32 | 51.9 | 83.2 | 25.6 | 54.7 | 56.2 | 86.2 | 8.7 | 27.2 | 47.4 | 80.8 |
| RN50+VAE | 65.1 | 90.3 | 19.8 | 47.1 | 65.2 | 90.7 | 5.9 | 18.4 | 43.6 | 76.8 |
| B32+VAE | 71.3 | 93.1 | 26.2 | 57.1 | 73.7 | 94.1 | 9.3 | 27.2 | 59.7 | 88.9 |
| RN50+B32+VAE | 72.6 | 93.5 | 27.4 | 58.9 | 75.7 | 94.6 | 9.1 | 28.4 | 62.5 | 90.0 |

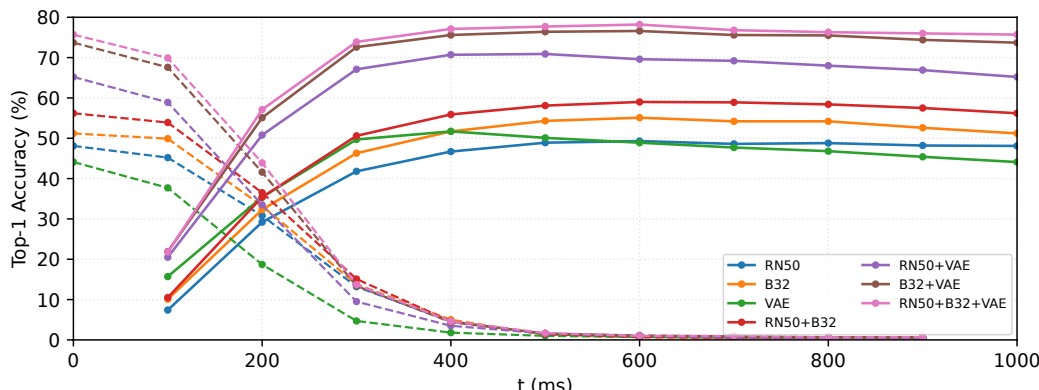

Figure 9: Top-1 retrieval accuracy (%) as a function of the EEG time window for different visual encoder configurations. Solid lines show cumulative windows $[0, t]$; dashed lines show tail windows $[t, 1000]$, with $t=0$ denoting the full $[0, 1000]$ ms interval.

time windows, the full RN50+B32+VAE fusion consistently achieves the best performance, further supporting the effectiveness of our approach.

**Encoder-masking analysis** To probe the relative contribution of each visual branch in the RN50+B32+VAE fusion model, we keep the trained model fixed and zero-mask individual encoder embeddings only at inference time (Tab. 16). Removing the VAE or B32 branches causes substantial drops in both intra- and inter-subject accuracy, whereas masking RN50 has a comparatively smaller effect, indicating that VAE and B32 provide the dominant complementary signals in this fusion configuration.

Table 16: Encoder-masking ablation for brain-to-image retrieval with the RN50+B32+VAE fusion setting. We report average top-1/top-5 accuracy (%) for 200-way zero-shot retrieval on THINGS-EEG, masking one visual encoder at inference under identical training configurations.

| Setting | Intra-subject | | Inter-subject | |
|---|---|---|---|---|
| | Top-1 | Top-5 | Top-1 | Top-5 |
| w/o RN50 | 70.6 | 92.7 | 19.8 | 41.2 |
| w/o B32 | 41.2 | 73.2 | 11.9 | 27.4 |
| w/o VAE | 29.6 | 60.2 | 9.3 | 23.4 |
| FULL | **75.7** | **94.6** | **20.0** | **44.1** |

