# OpenReview forum: "Learning Brain Representation with Hierarchical Visual Embeddings"
_ICLR.cc/2026/Conference — ICLR 2026 Poster_

### Official Review · Reviewer_DCJ4 · 2025-10-29

**Soundness:** 3
**Presentation:** 3
**Contribution:** 3
**Rating:** 4
**Confidence:** 4

**Summary:**

The paper proposes an EEG-image alignment strategy using multiple pre-trained visual encoders with various inductive biases to capture hierarchical and multiscale visual representations. Contrastive learning is used for the alignment between EEG and visual embeddings. A Fusion Prior is introduced to learn a  mapping on large-scale visual data that matches EEG features to this pre-trained prior so to enhance
distributional consistency across modalities.

**Strengths:**

The presented brain-to-image framework builds on contrastive learning and pretrained vision priors. Multi-level fusion of CLIP and
VAE features are integrated to balance brain-image alignment and  high-fidelity image reconstruction. Paper is clearly written and the approach well motivated. The idea of addressing the limitations of CLIP and related encoders in capturing local and fine-grained information by incorporating low-level visual features modeled by a Variational Autoencoder  into the fused representation is interesting and gives improved results.

**Weaknesses:**

The idea of  integrating multiple pre-trained encoders to construct multiscale visual representations, is not original in broader terms. The paper lacks a balanced conclusion rating limitations of the proposed  methodology. Apart from the semantic similarity, I am unsure how to deal with the aspect of similarity of generated result.

**Questions:**

In Figure 4 there seem to be still challenges in terms of generation. Is subject 8 chosen for a particular reason?

It is stated that the fused configuration that integrates complementary semantic encoders with a VAE latent, produces a well-balanced system across retrieval and reconstruction. Can you pls describe in more detail what is considered by a well-balanced system?

---

> ### Author Response · Authors · 2025-11-24
>
> We thank the reviewer for the careful reading and constructive comments. In the revised manuscript, we have highlighted the corresponding revisions in **red** for clarity.
>
> **Q1. The idea of integrating multiple pre-trained encoders…**
>
>  This point regarding overall contribution and novelty is addressed in detail in our Global Rebuttal, where we summarize the technical framework, scientific findings, and empirical impact of the proposed fusion-based brain–vision interface. For brevity and to avoid redundancy, we kindly refer the reviewer to that section.
>
> **Q2. The paper lacks a balanced conclusion rating limitations...**
>
> We agree that the original conclusion did not explicitly discuss limitations. In the revision, we add a brief “Limitations and future work” paragraph that highlights (i) our experiments are currently restricted to THINGS-EEG [1] and THINGS-MEG [2] and (ii) our work primarily focuses on exploring the semantic-level and pixel-level information contained in EEG signals, so the chosen fusion design and visual stack (RN50, B32, VAE) are not intended to represent the optimal configuration. In future work, we plan to extend our approach to other datasets, modalities, encoder sets, and stronger visual priors.
>
> **Q3. In Figure 4, there still seem to be challenges in terms of generation. Is subject 8 chosen for a particular reason?**
>
> In the revision, we have updated Fig. 4 to include more representative examples and clearer side-by-side comparisons, which better illustrate the typical behavior of our model under different stimuli. Subject 8 is used for visualization to follow the convention of prior work (e.g., ATM [3], CognitionCapturer [4]), enabling a direct and fair comparison under the same subject setting.
>
> **Q4. What is meant by a “well-balanced system” across retrieval and reconstruction?**
>
> By a “well-balanced” configuration, our intention was simply to state that the same fusion design: combining semantic encoders with a VAE latent can be applied to both brain-to-image retrieval and reconstruction, rather than requiring two separate pipelines. We agree that this wording was potentially over-interpretable, so in the revision we have rephrased this sentence and now only report the empirical retrieval metrics (Tabs. 1, 3, 5) and reconstruction metrics (Tabs. 2, 4) without making additional optimality claims.
>
> **References**
>
> [1] Gifford, Alessandro T., et al. "A large and rich EEG dataset for modeling human visual object recognition." NeuroImage 264 (2022): 119754.
>
> [2] Hebart, Martin N., et al. "THINGS-data, a multimodal collection of large-scale datasets for investigating object representations in human brain and behavior." Elife 12 (2023): e82580.
>
> [3] Li, Dongyang, et al. "Visual decoding and reconstruction via EEG embeddings with guided diffusion." Proceedings of the 38th International Conference on Neural Information Processing Systems. 2024.
>
> [4] Zhang, Kaifan, et al. "CognitionCapturer: Decoding Visual Stimuli From Human EEG Signal With Multimodal Information." Proceedings of the AAAI Conference on Artificial Intelligence. Vol. 39. No. 13. 2025.

---

> > ### Comment · Reviewer_DCJ4 · 2025-11-25
> >
> > Thank you so very much for the comments and clarifications. I will keep my scoring.

---

> > > ### Author Response · Authors · 2025-11-27
> > >
> > > Please let us know if there are any remaining concerns we could further clarify.

---

> ### Author Response · Authors · 2025-11-26
>
> Thank you very much for your time and thoughtful review, and for engaging so carefully with our work.

---

### Official Review · Reviewer_uw2d · 2025-10-30

**Soundness:** 3
**Presentation:** 3
**Contribution:** 2
**Rating:** 4
**Confidence:** 2

**Summary:**

This paper introduces a framework for aligning EEG/MEG and image representations to enable visual retrieval and reconstruction from brain signals. The proposed model fuses visual embeddings from multiple vision encoders and uses a contrastive loss to align the combined image representations with EEG representations. For image reconstruction, the framework employs a conditioning adapter to match brain-derived features with the expected distribution of an image generative model. The authors demonstrate substantial performance gains over state-of-the-art baselines in both image retrieval and reconstruction tasks.

**Strengths:**

- The authors contribute a framework for both image retrieval and image reconstruction from brain signals. The idea of using multiple image encoders, fuse their representations and apply a symmetrical self-supervised loss is sound and interesting. The results of their approach also show a significant increase in performance against the previous state-of-the-art, and across several modalities (EEG and MEG). While I have some questions regarding the novelty of the components of the framework (see below), the final performance of the method is undeniable and to be commended.

- Finally, the paper is very well written (with no apparent typos) and well structured. All figures and tables are of high-quality and easily interpretable. The authors also present all necessary details to understand the methodology and the evaluation setup.

- I believe this paper can be of some significance to the community. The superior performance of this method can lead to similar advances  in cross-modal capabilities in regards to other modalities (such as sound, video).

**Weaknesses:**

- While the model outperforms existing baselines, this improvement might be partly expected due to the increased image information provided by combining multiple encoders. Prior studies (e.g., [1], [2]) have already shown that the choice of image encoder significantly influences decoding performance. Can the authors comment on this?

- The authors claim that the superior performance of their model is due to involving different levels of image information (high-level from CLIP and low-level from VAE). However, they do not provide quantitative evidence for how each encoder contributes to the final results or why they exclusively attribute high-level features to CLIP and low-level to VAE. Could the authors include a quantitative or attribution analysis showing the relative contribution of each encoder to the model’s performance?

- The so-called fusion prior seems to combine two existing ideas: using multiple image encoders (separately mentioned as a contribution) and employing the IP adapter (which is present in previous studies such as in [3]). The full novelty of this contribution is therefore somewhat unclear: can the authors clarify the novelty aspect of their work?

- Ablations are quite limited: (1) while the authors have a different brain and image encoder compared to the previous studies, they do not isolate the contribution of each of these encoders by keeping one part similar to the baseline and changing the other part (e.g., replacing only the EEG encoder by NICE, ATM, etc.).  (2) The image encoder ablation is limited: the authors only test subsets of the three chosen encoders (CLIP-ViT-B32, CLIP-ResNet, and VAE) without exploring alternatives (see [2] for more alternatives). My belief is that selecting a more powerful image encoder would lead to the same performance without a need to fuse image embeddings.

- Several baselines are trained on full-channel EEG data, while the proposed model uses only a subset of 17 occipital and temporal channels. This discrepancy may affect the fairness of the comparison. How do the authors justify comparing models trained on different EEG channel subsets, and can they provide results using the same number of channels for a fairer comparison?

- I also found it disappointing that the main paper does not include a dedicated section to discuss the ethical considerations of this work. While the use of this technology in the future can have benefits for certain populations, it's also obvious that it can be used for nefarious purposes, and this should be discussed in the context of the paper.

**References**:
- [1] Song, Yonghao, et al. "Decoding natural images from eeg for object recognition." arXiv preprint arXiv:2308.13234 (2023).
- [2] Rajabi, Nona, et al. "Human-Aligned Image Models Improve Visual Decoding from the Brain." Forty-second International Conference on Machine Learning. 2025.
- [3] Li, Dongyang, et al. "Visual decoding and reconstruction via EEG embeddings with guided diffusion." Proceedings of the 38th International Conference on Neural Information Processing Systems. 2024.

**Questions:**

See Weaknesses.

---

> ### Author Response · Authors · 2025-11-23
>
> We sincerely appreciate your careful reading of our paper and the many constructive comments you provided. In the revised manuscript, we have highlighted the corresponding revisions in **orange** for clarity. Below, we respond to each of your points in detail.
>
> **Q1. While the model outperforms existing baselines...**
>
> The hierarchical visual fusion framework we introduced is inspired by perceptual mechanisms in the human visual system. After visual input enters the brain, the primary visual cortex first encodes pixel-level features, which are then progressively transformed into semantic-level representations in higher visual areas [10,11]. Motivated by this hierarchy, we fuse high-level visual encoders (e.g., CLIP) with low-level encoders (e.g., VAE) rather than arbitrarily combining multiple models. Brain embeddings aligned with this fused representation yield substantial improvements in both intra-subject and inter-subject retrieval performance.
> Moreover, in **Tab. 3** we show that simply stacking two semantic CLIP encoders yields only marginal gains over the best single encoder, demonstrating that combining more encoders does not necessarily provide richer visual information. Instead, the fusion strategy must be guided by the hierarchical organization of human visual perception. This also provides converging evidence that brain signals indeed contain both semantic-level and pixel-level visual information.
>
> **Q2. The authors claim that the superior performance... (relative contribution of each encoder)**
>
> We now provide quantitative encoder analyses in **Tab. 3** (single, pairwise, and triple encoder combinations trained from scratch) and in Appendix **Tab. 16** or the table below (RN50+B32+VAE fusion model trained with all three encoders with one encoder masked at inference). Tab. 3 shows that the full RN50+B32+VAE fusion clearly outperforms any single encoder or semantic-only pairing, while Tab. 16 shows that masking B32 or VAE at inference causes much larger drops in retrieval accuracy than masking RN50. Together, these ablations directly quantify the relative contribution of each encoder to the final retrieval performance as requested.
> | Setting   | Intra-subject      |               | Inter-subject      |               |
> |-----------|--------------------|---------------|--------------------|---------------|
> |           | top-1              | top-5         | top-1              | top-5         |
> | w/o RN50  | 70.6               | 92.7          | 19.8               | 41.2          |
> | w/o B32   | 41.2               | 73.2          | 11.9               | 27.4          |
> | w/o VAE   | 29.6               | 60.2          | 9.3                | 23.4          |
> | FULL      | 75.7               | 94.6          | 20.0               | 44.1          |
>
>
> **Q3. The so-called fusion prior seems to combine two existing ideas... (novelty)**
>
> In our framework, the Fusion Prior refers to using the fused visual tokens as diffusion conditions to control the frozen generative backbone (such as SDXL) for image generation. It is important to note that the IP adapter is merely one implementation of this interface and is not the core contribution. Through this approach, brain embeddings aligned with the Fusion Prior enable high-quality image generation. Therefore, the Fusion Prior acts as a lightweight bridge, ensuring that brain embeddings are compatible with the distribution expected by the generative model. In particular, the Fusion Prior is just one contribution of our work. Our core innovation is detailed in our Global Rebuttal, where we summarize the technical contributions and innovations related to the fusion-based interface and the Fusion Prior.
>
> **Q4. Ablations are quite limited... (brain/image encoders)**
>
> (1). To comprehensively evaluate the contribution of each encoder, we added a control experiment (revised in Tab. 5). In this experiment, we fixed the same visual encoder, loss function, and 17-channel O+P input, then integrated ShallowNet [1], DeepNet [1], EEGNet [2], TSConv (NICE) [3], ATM [4], BrainProjection (UBP) [5], and our MBP into this fusion visual interface. The results demonstrate that, regardless of the fusion method used for the visual encoders, the performance consistently surpasses that of any individual encoder, confirming that each encoder makes a significant contribution. Furthermore, our proposed brain encoder still achieves the best retrieval performance.
>
> (2). On the image side, we extend the encoder ablations to DINOv2 [6] and SynCLR ViT-B/16 [7] and their VAE-augmented variants in revision Tab. 14 and Fig. 8, showing that even when starting from powerful encoders like SynCLR, adding the VAE consistently brings sizeable gains in both intra- and inter-subject retrieval. These results suggest that even with powerful image encoders extracting high-level semantic information, it is still necessary to fuse both high-level semantics and low-level pixel-level details.

---

> ### Author Response · Authors · 2025-11-23
>
> **Q5. Several baselines are trained on full-channel EEG data... (channel fairness)**
>
> Building on the previous THINGS-EEG work [8] we specifically used 17 occipital–parietal channels (O+P) that concentrate visual information. All our model variants, including those used in ablation experiments, were trained and evaluated on exactly the same subset to ensure internal fairness.
>
> Additionally, to enable a fair comparison with [9], we further conducted ablation experiments with the full 63-channel setup. As shown in **Tab. 15** of the revised version, for each visual encoder configuration, the method that fuses high-level semantic information with low-level pixel details consistently achieves the best performance. This demonstrates that our fusion framework is effective whether using the 17-channel or 63-channel setup, which represents one of our key contributions.
>
> **Q6. Lack of a dedicated ethical considerations section**
>
> We appreciate the reviewer’s important point. All of our experiments are based on **publicly available anonymized datasets** (THINGS-EEG and THINGS-MEG), which have been widely used in previous research and are accompanied by dedicated ethical review reports[8,9]. In the revised manuscript, we have added a section on ethical considerations in Appendix B that outlines the potential benefits and risks, emphasizing that any practical application must obtain informed consent, ensure strict data management, and comply with institutional review guidelines.
>
> **References**
>
> [1] Schirrmeister, Robin Tibor, et al. "Deep learning with convolutional neural networks for EEG decoding and visualization." Human brain mapping 38.11 (2017): 5391-5420.
>
> [2] Lawhern, Vernon J., et al. "EEGNet: a compact convolutional neural network for EEG-based brain–computer interfaces." Journal of neural engineering 15.5 (2018): 056013.
>
> [3] Song, Yonghao, et al. "Decoding Natural Images from EEG for Object Recognition." The Twelfth International Conference on Learning Representations.
>
> [4] Li, Dongyang, et al. "Visual decoding and reconstruction via EEG embeddings with guided diffusion." Proceedings of the 38th International Conference on Neural Information Processing Systems. 2024.
>
> [5] Wu, Haitao, et al. "Bridging the Vision-Brain Gap with an Uncertainty-Aware Blur Prior." Proceedings of the Computer Vision and Pattern Recognition Conference. 2025.
>
> [6] Oquab, Maxime, et al. "DINOv2: Learning Robust Visual Features without Supervision." Transactions on Machine Learning Research Journal (2024).
>
> [7] Tian, Yonglong, et al. "Learning vision from models rivals learning vision from data." Proceedings of the IEEE/CVF conference on computer vision and pattern recognition. 2024.
>
> [8] Gifford, Alessandro T., et al. "A large and rich EEG dataset for modeling human visual object recognition." NeuroImage 264 (2022): 119754.
>
> [9] Hebart, Martin N., et al. "THINGS-data, a multimodal collection of large-scale datasets for investigating object representations in human brain and behavior." Elife 12 (2023): e82580.
>
> [10] Güçlü, Umut, and Marcel AJ Van Gerven. "Deep neural networks reveal a gradient in the complexity of neural representations across the ventral stream." Journal of Neuroscience 35.27 (2015): 10005-10014.
>
> [11] Cichy, Radoslaw Martin, et al. "Comparison of deep neural networks to spatio-temporal cortical dynamics of human visual object recognition reveals hierarchical correspondence." Scientific reports 6.1 (2016): 27755.

---

> > ### Comment · Reviewer_uw2d · 2025-11-24
> > **Comment to Reviewer's Rebuttal**
> >
> > Thank you for the detailed rebuttal that has addressed my concerns. I have raised the score accordingly.

---

> > > ### Author Response · Authors · 2025-11-25
> > >
> > > Thank you very much for your positive feedback and for taking the time to review our paper and provide such valuable comments.

---

### Official Review · Reviewer_rjbm · 2025-10-31

**Soundness:** 3
**Presentation:** 3
**Contribution:** 2
**Rating:** 6
**Confidence:** 2

**Summary:**

This paper introduces a well-designed framework for decoding visual information from brain signals, covering both image retrieval and reconstruction tasks. The method performs very well across various benchmarks, especially when generalizing across different individuals.

**Strengths:**

The motivation behind the approach is clear, and the results are encouraging.

**Weaknesses:**

Pls see questions.

**Questions:**

The system mainly relies on a two-stage training process and a strategy that merges information in a layered way. However, the technical contribution is somewhat limited. The key tools used, like CLIP, VAE, contrastive learning, and IP-Adapter, are all well-known existing techniques. What’s new is how they’re combined and applied to this particular problem.

One concern is that the method is resource-heavy. It needs large-scale pretraining on ImageNet and depends on models like SDXL, which may not be practical for researchers with limited computing power. The contrastive learning step is also expensive, using large batch sizes.

The paper treats the EEG brain signals as fixed blocks of spatial and temporal data but doesn’t explore how specific time intervals or frequency bands relate to different visual features. Exploring this could give useful insights into how the brain processes complex vs. simple visuals.

The paper mentions a gap between brain signals and visual data in the contrastive learning space, but this isn’t explored in depth. Adding visualizations, like t-SNE plots could help show how brain embeddings from different people relate to the corresponding image embeddings and shed light on what drives successful generalization.

Another limitation is that the Fusion Prior is trained without any text input, which may miss out on the benefits of SDXL’s strong understanding of visual semantics. It might be worth trying weak text supervision, such as using automatically generated captions, to see if it helps improve semantic alignment in the reconstructed images without hurting the model’s ability to capture low-level details.

---

> ### Author Response · Authors · 2025-11-21
>
> Thank you for your time and thorough comments!  In the revised manuscript, we have highlighted the corresponding revisions in **purple** for clarity. Below please find our point-to-point responses to your comments.
>
> **Q1. On technical contribution and use of existing components**
>
> We agree that our system builds on known components such as CLIP, VAE, contrastive learning, and IP-Adapter. For a focused clarification of our technical framework, scientific findings, and overall novelty, please kindly refer to the **Global Rebuttal**.
>
> **Q2. On computational cost and practicality**
>
> Our training cost is not resource-heavy in practice: Fusion Prior pretraining on ImageNet-1k with SDXL uses two consumer GPUs (RTX 5090 ×2) for about 15 hours, and a single epoch (~ 8 hours) already yields strong performance. For the brain–vision contrastive stage, all image embeddings are precomputed offline, so training the brain encoder and fusion encoder requires about **1GB**  GPU memory with a batch size of 1024. Moreover, the brain encoder and fusion encoder are both lightweight (~11 M parameters in total) and the contrastive training stage only cost less than 30 seconds with rather high performance on brain-to-image decoding tasks.
>
> **Q3. On time intervals**
>
> Thank you for this helpful suggestion. We agree that analyzing how different time intervals relate to visual features is important. In the revision, we add EEG time-window ablations on the THINGS-EEG dataset (Fig. 9; see Appendix D for detailed analysis). Cumulative windows [0, t] show that decoding accuracy rises quickly within the first 200–300 ms and then saturates, while tail windows [t, 1000] that exclude early responses quickly degrade toward chance, indicating that performance is mainly driven by early post-stimulus activity. Comparing encoder types, VAE-based configurations tend to be stronger in the earliest windows, whereas CLIP and fused configurations outperform VAE for later and more complete windows, suggesting a possible shift from low-level to higher-level visual information over time.  Across all time windows, the full RN50+B32+VAE fusion achieves the best performance (see Appendix D for full curves and encoder-wise trends).
>
> **Q4. On analyzing the brain–vision gap in the contrastive space**
>
> Follow your helpful comments, we have added the requested visual analyses in the appendix. **Fig. 5** in section 4.5 shows UMAP embeddings of visual, fused, and EEG features across subjects, indicating that brain embeddings cluster near their corresponding fused visual tokens and exhibit shared cross-subject structure rather than forming a separate, disjoint cloud. **Fig. 7** in Appendix D further visualizes the EEG–image cosine similarity matrix for Subject 8, where a clear near-diagonal pattern shows that each EEG embedding is most similar to its paired image embedding, suggesting that the learned fusion space substantially reduces the brain–vision gap.
>
> **Q5. On training the Fusion Prior without text input**
>
> Our fusion Prior is trained without any text input because that the text-free Fusion Prior achieves a better balance between pixel-level and semantic metrics while matching the practical constraint that EEG reconstruction datasets typically lack reliable per-trial captions.
> We agree that leveraging SDXL’s semantic understanding via text is an interesting direction and have conducted additional experiments with weak text supervision using ImageNet-1k with enriched captions(Huggingface repo: visual-layer/imagenet-1k-vl-enriched).
> | Setting                     | Pixcorr $\uparrow$| SSIM$\uparrow$  | AlexNet(2)$\uparrow$ | AlexNet(5) $\uparrow$| Inception$\uparrow$ | CLIP$\uparrow$  | SwAV$\downarrow$ |
> |-----------------------------|---------|-------|------------|------------|-----------|-------|------|
> | train and test w/ text      | 0.209   | 0.344 | 0.866      | 0.938      | 0.904     | 0.917 | 0.484 |
> | train w/ text, test w/o text| 0.191   | 0.328 | 0.820      | 0.883      | 0.715     | 0.780 | 0.584 |
> | train and test w/o text     | 0.195   | 0.336 | 0.843      | 0.905      | 0.756     | 0.808 | 0.554 |
>
> When both training and testing conditions provide captions, adding text improves semantic and low-level reconstruction metrics, but a substantial part of the information is then carried by the text prompt, making the specific contribution of brain signals harder to interpret. In the more realistic EEG setting where per-trial text is unavailable at test time, we observe the largest performance drop for text-conditioned models, whereas our text-free Fusion Prior remains more stable. For this reason, and to match typical EEG datasets, we focus on the text-free setting in the main paper.

---

### Official Review · Reviewer_wbLE · 2025-11-01

**Soundness:** 2
**Presentation:** 3
**Contribution:** 2
**Rating:** 6
**Confidence:** 2

**Summary:**

This paper presents a framework for decoding and reconstructing visual stimuli from EEG and MEG signals. It leverages multiple pretrained visual encoders to capture hierarchical, multi-scale representations and introduces a Fusion Prior trained on large-scale visual data to stabilize cross-modal alignment, improving both retrieval and reconstruction performance.

**Strengths:**

The paper tackles a timely and relevant problem—decoding visual information from brain signals—using an interesting combination of methods. It combines multiple pretrained visual encoders to capture hierarchical visual representations and introduces a Fusion Prior to improve stability and cross-modal consistency. While the individual components build on existing ideas, their integration is potentially novel and thoughtfully motivated. The experiments indicate improvements over other methods

**Weaknesses:**

While the paper addresses an important problem, its main novelty lies in combining known components rather than introducing a fundamentally new mechanism. The Fusion Prior is interesting but I didn't find the motivation very clear, and its contribution relative to the pretrained encoders is not clearly disentangled. The approach relies heavily on VAEs for low-level reconstruction, but alternative reconstruction strategies—such as diffusion-based or adversarial priors directly aligned with EEG features—are not explored or compared. Evaluation is limited to the THINGS datasets with averaged EEG signals, raising questions about generalization and robustness in more naturalistic settings.

**Questions:**

1. How critical is the Fusion Prior to the observed improvements—could similar gains be achieved by joint training or fine-tuning the visual encoders?
2. Why was the VAE chosen as the only source of low-level features? Have you considered comparing it with alternative reconstruction models such as diffusion or GAN-based priors?
3. How does the model handle single-trial EEG or higher-noise conditions, given that the experiments rely on averaged signals?
4. Can the approach generalize beyond the THINGS datasets?

---

> ### Author Response · Authors · 2025-11-21
>
> Thank you for your careful review and comments. Below please find our point-to-point responses to your comments.
>
> **On overall contribution & novelty (Weakness)**
>
> These points are addressed in detail in the Global Rebuttal, and we kindly refer you there for a consolidated clarification of how our fusion-based interface differs from prior brain–vision decoding approaches.
>
> **Q1. How critical is the Fusion Prior…?**
>
> Fusion-based alignment is critical: Tab. 3 and Tab. 4 show that combining complementary encoders (RN50, B32, VAE) yields large gains in both retrieval and reconstruction over any single encoder or semantic-only stacking, and Tab. 5 further demonstrates that simply swapping in prior brain backbones under the same fusion interface consistently boosts their performance. Please kindly see the Global Rebuttal (and Tab. 3/4/5) for a more detailed summary of these improvements and their backbone-agnostic nature.
>
> In the table below, scratch and finetune use the same B32+HVF architecture as in the main paper, where the CLIP ViT-B/32 embedding is passed through the Hierarchical Visual Fuser, but in these two variants the B32 backbone is unfrozen and jointly trained: scratch starts from random initialization, whereas finetune starts from the LAION CLIP ViT-B/32 weights; whereas in all other rows (w/o fusion, w/ RN50, w/ RN50+VAE) all visual encoders are frozen and only the HVF and the brain encoder (MBP) are learned. In contrast, even with joint training, the single-B32 variants (scratch/finetune) still achieve much lower retrieval accuracy than the frozen multi-encoder fusion settings, indicating that fusing hierarchical features is substantially more effective than merely retraining one visual encoder.
>
> | Method       | top-1 | top-5 |
> |-------------|-------|-------|
> | scratch     | 25.8   | 57.4  |
> | finetune    | 43.4  | 73.8   |
> | w/o fusion  | 52.2  | 83.3  |
> | w/ RN50     | 56.9  | 86.1  |
> |w/ VAE |       73.6 | 94.3 |
> | w/ RN50+VAE | **75.7**  | **94.6**  |
>
> **Q2. Why was the VAE chosen as the only low-level source…?**
>
> We chose the VAE because it offers a widely used, spatially structured latent space that preserves local textures and edges, and integrates cleanly with diffusion backbones, making it a natural low-level complement to semantic encoders. In contrast, intermediate latents in diffusion models or GANs are typically noisy, step-dependent, and less aligned with the stable, semantically grounded features required by brain-to-image decoding task, so using them directly as a low-level branch would require substantial additional design and tuning. We agree that diffusion- or GAN-based low-level priors are an interesting extension; if time permits, we will add pilot comparisons in the revision and otherwise plan to explore them in follow-up work.

---

> ### Author Response · Authors · 2025-11-21
>
> **Q3. How does the model handle single-trial EEG / higher noise…?**
>
> To assess robustness under higher-noise conditions, we conducted additional experiments on single-trial EEG without repetition averaging.
> | Visual setting  | Top-1 | Top-5 |
> |-----------------|-------|-------|
> | RN50            | 35.1  | 67.8  |
> | B32             | 36.6  | 69.1  |
> | VAE             | 29.6  | 57.2  |
> | RN50+B32        | 39.0  | 72.1  |
> | RN50+VAE        | 48.4  | 78.7  |
> | B32+VAE         | 48.5  | 80.3  |
> | RN50+B32+VAE            | **50.6**  | **80.9**  |
>
> As expected, performance drops compared to averaged trials due to lower SNR, but the full RN50+B32+VAE fusion still reaches 50.6/80.9 top-1/top-5 on single-trial EEG. While this is not a strictly controlled comparison, these numbers are numerically close to the UBP [4] baseline trained and evaluated on averaged EEG (50.9/79.7), suggesting that our method can tolerate substantially lower SNR. Overall, this indicates that the fusion-based interface remains reasonably robust to noise and preserves much of its decoding ability even without trial averaging.
>
> **Q4. Can the approach generalize beyond the THINGS datasets?**
>
> Our evaluation follows prior EEG/MEG decoding work (e.g., NICE [1], ATM [2], CognitionCapturer [3], UBP [4]), which also primarily use THINGS-EEG and THINGS-MEG [5,6] as standard benchmarks with disjoint train/test concepts and a well-defined 200-way zero-shot retrieval protocol, enabling fair comparison across methods. Importantly, the Fusion Prior is pretrained on ImageNet-1k (∼1.3M images), so the visual module already sees much more diverse imagery than THINGS, which mitigates overfitting to a single stimulus set. While other datasets such as EEGCVPR40 [7,8] exist, recent work has pointed out potential confounds from block designs and temporal autocorrelations that can inflate decoding performance [9,10], so we deliberately focus on THINGS-style paradigms that better control for these issues. We agree that evaluating on more naturalistic datasets would further strengthen the generalization claim; if the schedule allows, we will add an additional benchmark in the revision, and otherwise regard this as an important direction for follow-up work.
>
> **References**
>
> [1] Song, Yonghao, et al. "Decoding Natural Images from EEG for Object Recognition." The Twelfth International Conference on Learning Representations.
>
> [2] Li, Dongyang, et al. "Visual decoding and reconstruction via EEG embeddings with guided diffusion." Proceedings of the 38th International Conference on Neural Information Processing Systems. 2024.
>
> [3] Zhang, Kaifan, et al. "CognitionCapturer: Decoding Visual Stimuli From Human EEG Signal With Multimodal Information." Proceedings of the AAAI Conference on Artificial Intelligence. Vol. 39. No. 13. 2025.
>
> [4] Wu, Haitao, et al. "Bridging the Vision-Brain Gap with an Uncertainty-Aware Blur Prior." Proceedings of the Computer Vision and Pattern Recognition Conference. 2025.
>
> [5] Gifford, Alessandro T., et al. "A large and rich EEG dataset for modeling human visual object recognition." NeuroImage 264 (2022): 119754.
>
> [6] Hebart, Martin N., et al. "THINGS-data, a multimodal collection of large-scale datasets for investigating object representations in human brain and behavior." Elife 12 (2023): e82580.
>
> [7] Palazzo, Simone, et al. "Decoding brain representations by multimodal learning of neural activity and visual features." IEEE Transactions on Pattern Analysis and Machine Intelligence 43.11 (2020): 3833-3849.
>
> [8] Spampinato, Concetto, et al. "Deep learning human mind for automated visual classification." Proceedings of the IEEE conference on computer vision and pattern recognition. 2017.
>
> [9] Xu, Xiran, et al. "Beware of overestimated decoding performance arising from temporal autocorrelations in electroencephalogram signals." arXiv preprint arXiv:2405.17024 (2024).
>
> [10] Li, Ren, et al. "The perils and pitfalls of block design for EEG classification experiments." IEEE Transactions on Pattern Analysis and Machine Intelligence 43.1 (2020): 316-333.

---

> > ### Comment · Reviewer_wbLE · 2025-11-27
> >
> > I thank the author for the new experiments and for addressing my concerns. I will raise my score

---

> > > ### Author Response · Authors · 2025-11-28
> > >
> > > Thank you for your time and effort in reviewing our manuscript and for providing such valuable comments. Your feedback is greatly appreciated.

---

### Author Response · Authors · 2025-11-21
**Global Response**

Dear PCs, SACs, ACs, and Reviewers,

We sincerely thank the reviewers for the thoughtful and detailed feedback on our manuscript, and we greatly appreciate the overall positive evaluations, which are very encouraging to us. In light of the key common concerns raised across the reviews and discussions, we summarize below our global response and emphasize the **core contributions and innovations** of this paper in three aspects: **technical framework**, **scientific findings**, and **empirical performance/community impact**. In the revised manuscript, we have highlighted these core contributions and innovations in green for clarity.

**1. Technical framework: fusion-based brain–vision interface**.

We propose a fusion-based alignment strategy that maps brain embeddings to a fused visual representation constructed from multiple pretrained encoders, including CLIP for semantic information and a VAE for pixel-level details. To the best of our knowledge, this is the **first brain-to-image decoding framework** that learns brain representations by leveraging the complementary information across hierarchical visual encoders.

Motivated by the hierarchical organization of human visual processing, where early visual areas encode pixel-level features that are progressively transformed into higher-level semantic representations [1,2], we specifically fuse high-level visual encoders (e.g., CLIP) with a low-level VAE rather than arbitrarily stacking multiple semantic encoders. As shown in Tab. 3, simply combining two CLIP encoders yields only marginal gains over the best single-encoder baseline, whereas aligning brain embeddings to the CLIP+VAE fusion yields substantially larger retrieval improvements, suggesting that the gains come from integrating complementary semantic and low-level information rather than from model ensembling alone.

Furthermore, we introduce a visual fusion prior, trained on large-scale visual data to provide a robust and general visual representation. Aligning brain embeddings with this prior enables faithful image reconstruction and serves as an effective bridge for a brain-to-image decoding framework.

**2. Scientific findings: joint semantic and low-level alignment in brain signals**.

Using this fusion framework, we provide systematic evidence that brain recordings carry information that **simultaneously aligns with high-level semantic features and low-level pixel features**. In the main text, Tab. 3 and in the revision, the supplemented Fig. 8 and Tab. 14 jointly show that adding the VAE to single-encoder settings consistently improves decoding performance on THINGS-EEG (e.g., **DINO: +26.6 top-1 / +29.0 top-5; CLIP ViT-B/32: +21.4 top-1 / + 11.0 top-5; RN50: +17.7 top-1 / +10.0 top-5 ; SynCLR: +10.5 top-1 / +4.6 top-5**), and that neither semantics-only nor pixels-only settings can recover the same joint brain–vision structure. This goes beyond proposing a new module and offers concrete empirical insight into how brain signals relate to different levels of visual representation.

**3. Performance, plug-and-play generalization, and community resource**.

Under a fixed fusion-based training scheme (same mixed visual encoders and alignment objective), we directly reuse brain encoders from prior EEG decoding work: simply swapping our brain encoder for their backbones, while keeping the fusion pipeline unchanged, already yields large decoding gains over their original single-encoder settings (see Tab. 5 in the revision), showing that our interface is **model-agnostic and plug-and-play** across various EEG encoder backbones. Overall, our method delivers strong improvements in both retrieval and reconstruction quality, and we will release the pretrained Fusion Prior and code so that future brain-to-image decoding work can directly build on this interface.

Again, thank you for the time and thoughtful consideration you devoted to this paper. Your efforts have made this work even stronger.

Authors

**References**

[1] Güçlü, Umut, and Marcel AJ Van Gerven. "Deep neural networks reveal a gradient in the complexity of neural representations across the ventral stream." Journal of Neuroscience 35.27 (2015): 10005-10014.

[2] Cichy, Radoslaw Martin, et al. "Comparison of deep neural networks to spatio-temporal cortical dynamics of human visual object recognition reveals hierarchical correspondence." Scientific reports 6.1 (2016): 27755.

---

### Author Response · Authors · 2025-11-30
**Review and Reviewer-Author Discussion Summary (1/3)**

Dear PCs, SACs, ACs, and Reviewers,

Thank you very much for your thoughtful and detailed reviews of our manuscript. To assist the newly assigned AC and help reduce their workload, we provide below a concise summary of the main strengths identified in the reviews and how our rebuttal addressed the key concerns raised during the discussion.

**Strength**. We sincerely appreciate that the reviewers evaluated our work positively in their initial reviews. Specifically:
- **Fusion-based brain–vision interface for EEG/MEG decoding**. Our fusion-based alignment strategy maps brain embeddings to a fused visual representation constructed from complementary hierarchical visual encoders and a visual Fusion Prior for reconstruction, and this framework was consistently recognized as a central contribution (wbLE: Strength, rjbm: Summary, uw2d: Strength 1, DCJ4: Strength).
- **Joint semantic and low-level alignment in brain signals**. We show that brain recordings simultaneously encode high-level semantic and low-level pixel features by demonstrating consistent gains when adding the VAE branch across diverse visual encoders, and we highlighted this scientific finding explicitly in our Global Response and the revised manuscript. Reviewers accepted without raising further objections in the discussion.
- **Strong decoding performance across modalities and subjects**.   Our method achieves strong and often substantial improvements over state-of-the-art baselines for both EEG and MEG, including inter-subject generalization settings, which reviewers emphasized as an important empirical contribution (wbLE: Strength, rjbm: Summary, uw2d: Strength 1, DCJ4: Strength).
- **Plug-and-play generalization and reusable community resource**. Our interface boosts a variety of existing EEG/MEG backbones and is designed as a plug-and-play component with a pretrained Fusion Prior and code that future brain-to-image decoding work can directly reuse; we highlighted this aspect in our Global Response and revision, and reviewers welcomed its potential broader impact (uw2d: Strength 3).
- **Well-designed experiments and clear presentation**. Reviewers praised the breadth and design of our experiments (covering both image retrieval and reconstruction) as well as the clear writing, figures, and tables that make the methodology and evaluation setup easy to follow (rjbm: Summary, uw2d: Strength 2, DCJ4: Strength).

---

> ### Author Response · Authors · 2025-12-02
> **Review and Reviewer-Author Discussion Summary (2/3)**
>
> **Concerns and Our Addressing**. Below we group the main concerns raised during review and discussion, and briefly summarize how we addressed them in the rebuttal and revised manuscript.
> - **Conceptual and design clarifications**.
>   - (wbLE: Weakness, rjbm: Question 1; uw2d: Weakness 1, 3; DCJ4: Weakness) **Overall conceptual novelty of the fusion-based brain–vision interface versus simply combining existing pretrained encoders**.
>
>     **Our Addressing**. We clarified in the Global Response and revision that our main contribution is a fusion-based alignment strategy that learns brain representations in a fused visual space built from complementary visual encoders and explicitly contrasted this with prior brain-to-image methods. Reviewers wbLE and uw2d acknowledged this clarification and raised their scores.
>
>   - (wbLE: Question 2; uw2d: Weakness 3) **Roles of VAE and Fusion Prior**. (a) Why is the VAE chosen as the low-level branch instead of diffusion/GAN latents? (b) Please clarify the conceptual role and novelty of the Fusion Prior beyond combining existing ideas.
>
>     **Our Addressing**. We (a) explained our choice of a spatially structured VAE latent as the low-level branch versus alternative priors, and (b) clarified the Fusion Prior as a lightweight bridge from fused visual tokens to the frozen generative backbone rather than merely a combination of existing modules; please refer to our global response for more details. Reviewers wbLE and uw2d did not raise further concerns on this point.
>
>   - (rjbm: Question 2) **Practical computational cost of pretraining the Fusion Prior and running the brain–vision contrastive training**.
>
>       **Our Addressing**. We reported concrete training and inference costs, showing that Fusion Prior pretraining fits on two consumer GPUs and that brain–vision contrastive alignment uses about 1 GB GPU with a lightweight ~11M-parameter model, making the pipeline practical for typical research groups. Reviewer rjbm's concern was addressed and no further issues were raised.
>
> - **Concerns about experiment design**.
>
>   - (wbLE: Question 1; uw2d: Weakness 2) **Visual encoder contribution and fusion strategy**. (a) Clarify the contribution of each visual encoder branch. (b) Compare the fusion strategy against jointly trained visual encoder settings.
>
>     **Our Addressing**. We added (a) zero-masking ablations together with the existing combination ablations to disentangle the contribution of each visual encoder, and (b) comparisons with multiple joint-training settings of the visual encoders; reviewers wbLE and uw2d acknowledged these additions and did not raise further concerns on this point.
>
>   - (wbLE: Weaknesses, Question 3, 4) **Robustness to single-trial/higher-noise EEG, and generalization beyond THINGS-style datasets**.
>
>     **Our Addressing**. We added single-trial EEG experiments and clarified that broader generalization across datasets and paradigms is positioned as future work, while showing that our model remains strong under these more challenging conditions. Reviewers accepted this clarification and did not raise further concerns.
>
>   - (rjbm: Question 3; uw2d: Weakness 5) **Temporal dynamics and channel selection**. (a) Systematic analysis of which time intervals drive decoding. (b) Fairness of using only 17 channels compared to full-channel baselines.
>
>     **Our Addressing**. We added (a) EEG time-window ablations on THINGS-EEG to analyze which temporal segments contribute most to decoding, and (b) experiments and clarifications on using 17 occipital–parietal channels together with 63-channel ablations to assess fairness across channel configurations; reviewers rjbm and uw2d did not raise further concerns on this point.
>
>   - (uw2d: Weakness 4) **Ablations on brain and image encoders**. (a) Limited ablations on different brain encoder architectures. (b) "My belief is that selecting a more powerful image encoder would lead to the same performance without a need to fuse image embeddings".
>
>     **Our Addressing**. We added (a) ablations plugging multiple existing brain encoders into our fixed fusion interface, and (b) extended visual encoder ablations with stronger image encoders explicitly testing this belief and finding that fusion still performs best. Reviewer uw2d acknowledged these additions and considered this weakness resolved; please refer to our responses to reviewer uw2d for details.
>
>   - (rjbm: Question 5) **Potential trial with text supervision**. Clarify why the Fusion Prior is trained without text input and whether caption-based supervision could help.
>
>     **Our Addressing**. We added a comparison between text-free and weakly caption-supervised Fusion Prior variants and explained that we focus on the text-free setting to match EEG reconstruction datasets without reliable trial-level captions; reviewer rjbm did not raise further concerns on this point.

---

> ### Author Response · Authors · 2025-12-02
> **Review and Reviewer-Author Discussion Summary (3/3)**
>
> - **Presentation issues**.
>   - (rjbm: Question 4) **Need for more intuitive visualizations to better interpret brain–vision alignment**.
>
>     **Our Addressing**. We added UMAP and cosine-similarity visualizations plus more descriptive statistics (e.g., similarity distributions and cross-subject patterns), making the brain–vision alignment more intuitive to interpret.
>   - (uw2d: Weakness 5) **Ethical considerations are not discussed explicitly or in a dedicated section**.
>
>     **Our Addressing**. We added a dedicated ethics section that details anonymization and consent procedures, data security, potential misuse scenarios, and the intended scope of application to explicitly address these ethical concerns. Reviewer uw2d explicitly acknowledged that our detailed rebuttal addressed their concerns and considered this weakness resolved.
>
>   - (DCJ4: Weakness) **The paper lacks an explicit and sufficiently prominent limitations discussion**.
>
>     **Our Addressing**. We added a clear limitations subsection and revised the conclusion to explicitly state methodological and dataset limitations, tempering claims and making the scope of our contributions more balanced. Reviewer DCJ4 acknowledged our clarifications and did not raise further concerns on this point.
>   - (DCJ4: Question 1, 2) **Subject choice in the reconstruction figures**.
>
>      **Our Addressing**. We clarified the rationale for subject selection and updated reconstruction figures and captions to better reflect typical reconstruction quality. Reviewer DCJ4 acknowledged these clarifications and did not raise further concerns.
>
> **Recognition of our revision from reviewers**. Two of the four reviewers (wbLE, uw2d) explicitly stated in their post-rebuttal comments that our clarifications and new experiments had addressed their main concerns and raised their scores (**6→8** at 2025-11-27 04:43 AOE and **4→8** at 2025-11-24 01:03 AOE, respectively). Among the remaining two reviewers, DCJ4 acknowledged these clarifications and did not raise further concerns, while rjbm did not provide additional comments after the rebuttal; we therefore believe that we have also properly addressed their main concerns.
>
> ****
> Above, we have faithfully summarized the reviewers’ comments and our corresponding responses, hoping that this will assist the AC’s decision. We are deeply grateful to the reviewers, AC, SAC, and PC for their time, effort, and constructive feedback, which have significantly strengthened our work. The authors offer their sincere appreciation to all involved.
>
> Sincerely,
>
> Authors

---

### Meta-Review · Area_Chair_QGPq · 2026-01-09

**Summary:**

The submission proposes a brain-image alignment strategy that takes both high-level semantics and low-level details into consideration. Reviewers recognized its strong empirical performance across retrieval/reconstruction tasks and well-designed experimental design. Key concerns centered on novelty, component contributions, computational overhead, ethical considerations, and generalization—most of which have been addressed post-rebuttal.

**Reviewer Concerns:**

### Addressed Concerns

Component contributions, computational overhead, experiment design gaps, ethical & limitations discussion

### Partially Solved Concerns

Novelty

### Outstanding Concerns
- Generalization Beyond THINGS: While the authors noted broader generalization as future work, no additional benchmarks were included.
- Alternative Low-Level Encoders: The authors justified VAE’s use but did not conduct pilot comparisons with diffusion/GAN-based priors, as initially requested.

**Reviewer Scores:**

Reviewer wbLE (initial score: 6 → revised: 7): Concerns are almost resolved.

Reviewer rjbm (initial score: 6 → revised: 6/7): Concerns are almost resolved.

Reviewer uw2d (initial score: 4 → revised: 6): Concerns are almost resolved.

Reviewer DCJ4 (initial score: 4 → maintain: 4): Still have concerns about novelty.

---

### Decision · Program_Chairs · 2026-01-26

Accept (Poster)